# SEQUENTIAL ATTENTION FOR FEATURE SELECTION

**Taisuke Yasuda**[*][†]
Carnegie Mellon University
taisukey@cs.cmu.edu

**MohammadHossein Bateni, Lin Chen, Matthew Fahrbach, Gang Fu,[*] and Vahab Mirrokni**
Google Research
{bateni,linche,fahrbach,thomasfu,mirrokni}@google.com

## ABSTRACT

Feature selection is the problem of selecting a subset of features for a machine learning model that maximizes model quality subject to a budget constraint. For neural networks, prior methods, including those based on $\ell_1$ regularization, attention, and other techniques, typically select the entire feature subset in one evaluation round, ignoring the residual value of features during selection, i.e., the marginal contribution of a feature given that other features have already been selected. We propose a feature selection algorithm called Sequential Attention that achieves state-of-the-art empirical results for neural networks. This algorithm is based on an efficient one-pass implementation of greedy forward selection and uses attention weights at each step as a proxy for feature importance. We give theoretical insights into our algorithm for linear regression by showing that an adaptation to this setting is equivalent to the classical Orthogonal Matching Pursuit (OMP) algorithm, and thus inherits all of its provable guarantees. Our theoretical and empirical analyses offer new explanations towards the effectiveness of attention and its connections to overparameterization, which may be of independent interest.

## 1 INTRODUCTION

Feature selection is a classic problem in machine learning and statistics where one is asked to find a subset of $k$ features from a larger set of $d$ features, such that the prediction quality of the model trained using the subset of features is maximized. Finding a small and high-quality feature subset is desirable for many reasons: improving model interpretability, reducing inference latency, decreasing model size, regularization, and removing redundant or noisy features to improve generalization. We direct the reader to Li et al. (2017b) for a survey on the role of feature selection in machine learning.

The widespread success of deep learning has prompted an intense study of feature selection algorithms for neural networks, especially in the supervised setting. While many methods have been proposed, we focus on a line of work that studies the use of *attention for feature selection*. The attention mechanism in machine learning roughly refers to applying a trainable softmax mask to a given layer. This allows the model to "focus" on certain important signals during training. Attention has recently led to major breakthroughs in computer vision, natural language processing, and several other areas of machine learning (Vaswani et al., 2017). For feature selection, the works of Wang et al. (2014); Gui et al. (2019); Skrlj et al. (2020); Wojtas & Chen (2020); Liao et al. (2021) all present new approaches for feature attribution, ranking, and selection that are inspired by attention.

One problem with naively using attention for feature selection is that it can ignore the *residual values* of features, i.e., the marginal contribution a feature has on the loss conditioned on previously-selected features being in the model. This can lead to several problems such as selecting redundant features or ignoring features that are uninformative in isolation but valuable in the presence of others.

---

[*]Corresponding authors
[†]This work was done while T.Y. was an intern at Google Research.

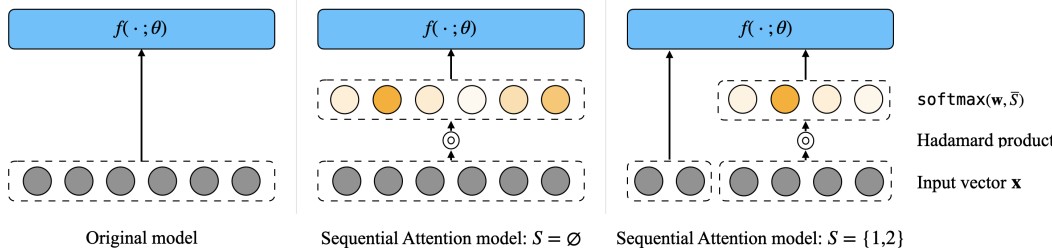

Figure 1: Sequential attention applied to model $f(\cdot; \boldsymbol{\theta})$. At each step, the selected features $i \in S$ are used as direct inputs to the model and the unselected features $i \notin S$ are downscaled by the scalar value $\mathrm{softmax}_i(\mathbf{w}, \overline{S})$, where $\mathbf{w} \in \mathbb{R}^d$ is the vector of learned attention weights and $\overline{S} = [d] \setminus S$.

This work introduces the *Sequential Attention* algorithm for supervised feature selection. Our algorithm addresses the shortcomings above by using attention-based selection *adaptively* over multiple rounds. Further, Sequential Attention simplifies earlier attention-based approaches by directly training one global feature mask instead of aggregating many instance-wise feature masks. This technique reduces the overhead of our algorithm, eliminates the toil of tuning unnecessary hyperparameters, *works directly with any differentiable model architecture*, and offers an efficient streaming implementation. Empirically, Sequential Attention achieves state-of-the-art feature selection results for neural networks on standard benchmarks. The code for our algorithm and experiments is publicly available.[1]

**Sequential Attention.** Our starting point for Sequential Attention is the well-known greedy forward selection algorithm, which repeatedly selects the feature with the *largest marginal improvement* in model loss when added to the set of currently selected features (see, e.g., Das & Kempe (2011) and Elenberg et al. (2018)). Greedy forward selection is known to select high-quality features, but requires training $O(kd)$ models and is thus impractical for many modern machine learning problems. To reduce this cost, one natural idea is to only train $k$ models, where the model trained in each step approximates the marginal gains of all $O(d)$ unselected features. Said another way, we can relax the greedy algorithm to fractionally consider all $O(d)$ feature candidates simultaneously rather than computing their exact marginal gains one-by-one with separate models. We implement this idea by introducing a new set of trainable variables $\mathbf{w} \in \mathbb{R}^d$ that represent *feature importance*, or *attention logits*. In each step, we select the feature with maximum importance and add it to the selected set. To ensure the score-augmented models (1) have differentiable architectures and (2) are encouraged to hone in on the best unselected feature, we take the *softmax* of the importance scores and multiply each input feature value by its corresponding softmax value as illustrated in Figure 1.

Formally, given a dataset $\mathbf{X} \in \mathbb{R}^{n \times d}$ represented as a matrix with $n$ rows of examples and $d$ feature columns, suppose we want to select $k$ features. Let $f(\cdot; \boldsymbol{\theta})$ be a differentiable model, e.g., a neural network, that outputs the predictions $f(\mathbf{X}; \boldsymbol{\theta})$. Let $\mathbf{y} \in \mathbb{R}^n$ be the labels, $\ell(f(\mathbf{X}; \boldsymbol{\theta}), \mathbf{y})$ be the loss between the model's predictions and the labels, and $\circ$ be the Hadamard product. Sequential Attention outputs a subset $S \subseteq [d] := \{1, 2, \ldots, d\}$ of $k$ feature indices, and is presented below in Algorithm 1.

**Theoretical guarantees.** We give provable guarantees for Sequential Attention for least squares linear regression by analyzing a variant of the algorithm called *regularized linear Sequential Attention*. This variant (1) uses Hadamard product overparameterization directly between the attention weights and feature values without normalizing the attention weights via $\mathrm{softmax}(\mathbf{w}, \overline{S})$, and (2) adds $\ell_2$ regularization to the objective, hence the "linear" and "regularized" terms. Note that $\ell_2$ regularization, or *weight decay*, is common practice when using gradient-based optimizers (Tibshirani, 2021). We give theoretical and empirical evidence that replacing the $\mathrm{softmax}$ by different overparameterization schemes leads to similar results (Section 4.2) while offering more tractable analysis. In particular, our main result shows that regularized linear Sequential Attention has the same provable guarantees as the celebrated *Orthogonal Matching Pursuit* (OMP) algorithm of Pati et al. (1993) for sparse linear regression, without making any assumptions on the design matrix or response vector.

**Theorem 1.1.** *For linear regression, regularized linear Sequential Attention is equivalent to OMP.*

---

[1]The code is available at: github.com/google-research/google-research/tree/master/sequential_attention

---

**Algorithm 1** Sequential Attention for feature selection.

---

1: **function** SEQUENTIALATTENTION(dataset $\mathbf{X} \in \mathbb{R}^{n \times d}$, labels $\mathbf{y} \in \mathbb{R}^n$, model $f$, loss $\ell$, size $k$)
2:      Initialize $S \leftarrow \varnothing$
3:      **for** $t = 1$ to $k$ **do**
4:          Let $(\boldsymbol{\theta}^*, \mathbf{w}^*) \leftarrow \arg\min_{\boldsymbol{\theta}, \mathbf{w}} \ell(f(\mathbf{X} \circ \mathbf{W}; \boldsymbol{\theta}), \mathbf{y})$, where $\mathbf{W} = \mathbf{1}_n \mathrm{softmax}(\mathbf{w}, \overline{S})^\top$ for

$$\mathrm{softmax}_i(\mathbf{w}, \overline{S}) := \begin{cases} 1 & \text{if } i \in S \\ \dfrac{\exp(\mathbf{w}_i)}{\sum_{j \in \overline{S}} \exp(\mathbf{w}_j)} & \text{if } i \in \overline{S} := [d] \setminus S \end{cases} \quad (1)$$

5:          Set $i^* \leftarrow \arg\max_{i \notin S} \mathbf{w}_i^*$         ▷ unselected feature with largest attention weight
6:          Update $S \leftarrow S \cup \{i^*\}$
7:      **return** $S$

---

We prove this equivalence using a novel two-step argument. First, we show that regularized linear Sequential Attention is equivalent to a greedy version of LASSO (Tibshirani, 1996), which Luo & Chen (2014) call *Sequential LASSO*. Prior to our work, however, Sequential LASSO was only analyzed in a restricted "sparse signal plus noise" setting, offering limited insight into its success in practice. Second, we prove that Sequential LASSO is equivalent to OMP in the fully general setting for linear regression by analyzing the geometry of the associated polyhedra. This ultimately allows us to transfer the guarantees of OMP to Sequential Attention.

**Theorem 1.2.** *For linear regression, Sequential LASSO (Luo & Chen, 2014) is equivalent to OMP.*

We present the full argument for our results in Section 3. This analysis takes significant steps towards explaining the success of attention in feature selection and the various theoretical phenomena at play.

**Towards understanding attention.** An important property of OMP is that it provably approximates the marginal gains of features—Das & Kempe (2011) showed that for any subset of features, the gradient of the least squares loss at its sparse minimizer approximates the marginal gains up to a factor that depends on the *sparse condition numbers* of the design matrix. This suggests that Sequential Attention could also approximate some notion of the marginal gains for more sophisticated models when selecting the next-best feature. We observe this phenomenon empirically in our marginal gain experiments in Appendix B.6. These results also help refine the widely-assumed conjecture that attention weights correlate with feature importances by specifying an exact measure of "importance" at play. Since a countless number of feature importance definitions are used in practice, it is important to understand which best explains how the attention mechanism works.

**Connections to overparameterization.** In our analysis of regularized linear Sequential Attention for linear regression, we do not use the presence of the softmax in the attention mechanism—rather, the crucial ingredient in our analysis is the Hadamard product parameterization of the learned weights. We conjecture that the empirical success of attention-based feature selection is primarily due to the explicit overparameterization.[2] Indeed, our experiments in Section 4.2 verify this claim by showing that if we substitute the softmax in Sequential Attention with a number of different (normalized) overparamterized expressions, we achieve nearly identical performance. This line of reasoning is also supported in the recent work of Ye et al. (2021), who claim that attention largely owes its success to the "smoother and stable [loss] landscapes" induced by Hadamard product overparameterization.

## 1.1 RELATED WORK

Here we discuss recent advances in supervised feature selection for deep neural networks (DNNs) that are the most related to our empirical results. In particular, we omit a discussion of a large body of works on unsupervised feature selection (Zou et al., 2015; Altschuler et al., 2016; Balın et al., 2019).

---

[2]Note that overparameterization here refers to the addition of $d$ trainable variables in the Hadamard product overparameterization, not the other use of the term that refers to the use of a massive number of parameters in neural networks, e.g., in Bubeck & Sellke (2021).

The *group LASSO* method has been applied to DNNs to achieve structured sparsity by pruning neurons (Alvarez & Salzmann, 2016) and even filters or channels in convolutional neural networks (Lebedev & Lempitsky, 2016; Wen et al., 2016; Li et al., 2017a). It has also be applied for feature selection (Zhao et al., 2015; Li et al., 2016; Scardapane et al., 2017; Lemhadri et al., 2021).

While the LASSO is the most widely-used method for relaxing the $\ell_0$ sparsity constraint in feature selection, several recent works have proposed new relaxations based on *stochastic gates* (Srinivas et al., 2017; Louizos et al., 2018; Balın et al., 2019; Trelin & Procházka, 2020; Yamada et al., 2020). This approach introduces (learnable) Bernoulli random variables for each feature during training, and minimizes the expected loss over realizations of the 0-1 variables (accepting or rejecting features).

There are several other recent approaches for DNN feature selection. Roy et al. (2015) explore using the magnitudes of weights in the first hidden layer to select features. Lu et al. (2018) designed the DeepPINK architecture, extending the idea of *knockoffs* (Benjamini et al., 2001) to neural networks. Here, each feature competes with a "knockoff" version of the original feature; if the knockoff wins, the feature is removed. Borisov et al. (2019) introduced the *CancelOut* layer, which suppresses irrelevant features via independent per-feature activation functions that act as (soft) bitmasks.

In contrast to these differentiable approaches, the combinatorial optimization literature is rich with greedy algorithms that have applications in machine learning (Zadeh et al., 2017; Fahrbach et al., 2019b;a; Chen et al., 2021; Halabi et al., 2022; Bilmes, 2022). In fact, most influential feature selection algorithms from this literature are sequential, e.g., greedy forward and backward selection (Ye & Sun, 2018; Das et al., 2022), Orthogonal Matching Pursuit (Pati et al., 1993), and several information-theoretic methods (Fleuret, 2004; Ding & Peng, 2005; Bennasar et al., 2015). These approaches, however, are not normally tailored to neural networks, and can suffer from quality, efficiency, or both.

Lastly, this paper studies *global* feature selection, i.e., selecting the same subset of features across all training examples, whereas many works consider *local* (or instance-wise) feature selection. This problem is more related to model interpretability, and is better known as *feature attribution* or *saliency maps*. These methods naturally lead to global feature selection methods by aggregating their instance-wise scores (Cancela et al., 2020). Instance-wise feature selection has been explored using a variety of techniques, including gradients (Smilkov et al., 2017; Sundararajan et al., 2017; Srinivas & Fleuret, 2019), attention (Arik & Pfister, 2021; Ye et al., 2021), mutual information (Chen et al., 2018), and Shapley values from cooperative game theory (Lundberg & Lee, 2017).

## 2 PRELIMINARIES

Before discussing our theoretical guarantees for Sequential Attention in Section 3, we present several known results about feature selection for linear regression, also called *sparse linear regression*. Recall that in the least squares linear regression problem, we have

$$\ell(f(\mathbf{X}; \boldsymbol{\theta}), \mathbf{y}) = \|f(\mathbf{X}; \boldsymbol{\theta}) - \mathbf{y}\|_2^2 = \|\mathbf{X}\boldsymbol{\theta} - \mathbf{y}\|_2^2. \tag{2}$$

We work in the most challenging setting for obtaining relative error guarantees for this objective by making *no distributional assumptions* on $\mathbf{X} \in \mathbb{R}^{n \times d}$, i.e., we seek $\tilde{\boldsymbol{\theta}} \in \mathbb{R}^d$ such that

$$\|\mathbf{X}\tilde{\boldsymbol{\theta}} - \mathbf{y}\|_2^2 \leq \kappa \min_{\boldsymbol{\theta} \in \mathbb{R}^d} \|\mathbf{X}\boldsymbol{\theta} - \mathbf{y}\|_2^2, \tag{3}$$

for some $\kappa = \kappa(\mathbf{X}) > 0$, where $\mathbf{X}$ is not assumed to follow any particular input distribution. This is far more applicable in practice than assuming the entries of $\mathbf{X}$ are i.i.d. Gaussian. In large-scale applications, the number of examples $n$ often greatly exceeds the number of features $d$, resulting in an optimal loss that is nonzero. Thus, we focus on the *overdetermined* regime and refer to Price et al. (2022) for an excellent discussion on the long history of this problem.

**Notation.** Let $\mathbf{X} \in \mathbb{R}^{n \times d}$ be the design matrix with $\ell_2$ unit columns and let $\mathbf{y} \in \mathbb{R}^n$ be the response vector, also assumed to be an $\ell_2$ unit vector.[3] For $S \subseteq [d]$, let $\mathbf{X}_S$ denote the $n \times |S|$ matrix consisting of the columns of $\mathbf{X}$ indexed by $S$. For singleton sets $S = \{j\}$, we write $\mathbf{X}_j$ for $\mathbf{X}_{\{j\}}$. Let $\mathbf{P}_S \coloneqq \mathbf{X}_S \mathbf{X}_S^+$ denote the projection matrix onto the column span $\mathrm{colspan}(\mathbf{X}_S)$ of $\mathbf{X}_S$, where $\mathbf{X}_S^+$ denotes the pseudoinverse of $\mathbf{X}_S$. Let $\mathbf{P}_S^\perp = \mathbf{I}_n - \mathbf{P}_S$ denote the projection matrix onto the orthogonal complement of $\mathrm{colspan}(\mathbf{X}_S)$.

---

[3]These assumptions are without loss of generality by scaling.

**Feature selection algorithms for linear regression.** Perhaps the most natural algorithm for sparse linear regression is greedy forward selection, which was shown to have guarantees of the form of (3) in the breakthrough works of Das & Kempe (2011); Elenberg et al. (2018), where $\kappa = \kappa(\mathbf{X})$ depends on *sparse condition numbers* of $\mathbf{X}$, i.e., the spectrum of $\mathbf{X}$ restricted to a subset of its columns. Greedy forward selection can be expensive in practice, but these works also prove analogous guarantees for the more efficient Orthogonal Matching Pursuit algorithm, which we present formally in Algorithm 2.

---

**Algorithm 2** Orthogonal Matching Pursuit (Pati et al., 1993).

---

1: **function** OMP(design matrix $\mathbf{X} \in \mathbb{R}^{n \times d}$, response $\mathbf{y} \in \mathbb{R}^n$, size constraint $k$)
2:     Initialize $S \leftarrow \varnothing$
3:     **for** $t = 1$ to $k$ **do**
4:         Set $\boldsymbol{\beta}_S^* \leftarrow \arg\min_{\boldsymbol{\beta} \in \mathbb{R}^S} \|\mathbf{X}_S \boldsymbol{\beta} - \mathbf{y}\|_2^2$
5:         Let $i^* \notin S$ maximize                ▷ maximum correlation with residual

$$\langle \mathbf{X}_i, \mathbf{y} - \mathbf{X}_S \boldsymbol{\beta}_S^* \rangle^2 = \langle \mathbf{X}_i, \mathbf{y} - \mathbf{P}_S \mathbf{y} \rangle^2 = \langle \mathbf{X}_i, \mathbf{P}_S^\perp \mathbf{y} \rangle^2$$

6:         Update $S \leftarrow S \cup \{i^*\}$
7:     **return** $S$

---

The LASSO algorithm (Tibshirani, 1996) is another popular feature selection method, which simply adds $\ell_1$-regularization to the objective in Equation (2). Theoretical guarantees for LASSO are known in the *underdetermined* regime (Donoho & Elad, 2003; Candes & Tao, 2006), but it is an open problem whether LASSO has the guarantees of Equation (3). Sequential LASSO is a related algorithm that uses LASSO to select features one by one. Luo & Chen (2014) analyzed this algorithm in a specific parameter regime, but until our work, *no relative error guarantees were known in full generality (e.g., the overdetermined regime)*. We present the Sequential LASSO in Algorithm 3.

---

**Algorithm 3** Sequential LASSO (Luo & Chen, 2014).

---

1: **function** SEQUENTIALLASSO(design matrix $\mathbf{X} \in \mathbb{R}^{n \times d}$, response $\mathbf{y} \in \mathbb{R}^n$, size constraint $k$)
2:     Initialize $S \leftarrow \varnothing$
3:     **for** $t = 1$ to $k$ **do**
4:         Let $\boldsymbol{\beta}^*(\lambda, S)$ denote the optimal solution to

$$\arg\min_{\boldsymbol{\beta} \in \mathbb{R}^d} \frac{1}{2}\|\mathbf{X}\boldsymbol{\beta} - \mathbf{y}\|_2^2 + \lambda\|\boldsymbol{\beta}_{\overline{S}}\|_1 \tag{4}$$

5:         Set $\lambda^*(S) \leftarrow \sup\{\lambda > 0 : \boldsymbol{\beta}^*(\lambda, S)_{\overline{S}} \neq \mathbf{0}\}$    ▷ largest $\lambda$ with nonzero LASSO on $\overline{S}$
6:         Let $A(S) = \lim_{\varepsilon \to 0}\{i \in \overline{S} : \boldsymbol{\beta}^*(\lambda^* - \varepsilon, S)_i \neq 0\}$
7:         Select any $i^* \in A(S)$                ▷ non-empty by Lemma 3.5
8:         Update $S \leftarrow S \cup \{i^*\}$
9:     **return** $S$

---

Note that Sequential LASSO as stated requires a search for the optimal $\lambda^*$ in each step. In practice, $\lambda$ can simply be set to a large enough value to obtain similar results, since beyond a critical value of $\lambda$, the feature ranking according to LASSO coefficients does not change (Efron et al., 2004).

## 3   EQUIVALENCE FOR LEAST SQUARES: OMP AND SEQUENTIAL ATTENTION

In this section, we show that the following algorithms are equivalent for least squares linear regression: regularized linear Sequential Attention, Sequential LASSO, and Orthogonal Matching Pursuit.

### 3.1   REGULARIZED LINEAR SEQUENTIAL ATTENTION AND SEQUENTIAL LASSO

We start by formalizing a modification to Sequential Attention that admits provable guarantees.

**Definition 3.1** (Regularized linear Sequential Attention). Let $S \subseteq [d]$ be the set of currently selected features. We define the *regularized linear Sequential Attention* objective by removing the

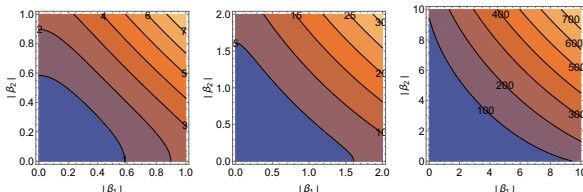

Figure 2: Contour plot of $Q^*(\boldsymbol{\beta} \circ \boldsymbol{\beta})$ for $\boldsymbol{\beta} \in \mathbb{R}^2$ at different zoom-levels of $|\boldsymbol{\beta}_i|$.

$\mathrm{softmax}(\mathbf{w}, \overline{S})$ normalization in Algorithm 1 and introducing $\ell_2$ regularization on the importance weights $\mathbf{w} \in \mathbb{R}^{\overline{S}}$ and model parameters $\boldsymbol{\theta} \in \mathbb{R}^d$ restricted to $\overline{S}$. That is, we consider the objective

$$\min_{\mathbf{w} \in \mathbb{R}^d, \boldsymbol{\theta} \in \mathbb{R}^d} \|\mathbf{X}(\mathbf{s}(\mathbf{w}) \circ \boldsymbol{\theta}) - \mathbf{y}\|_2^2 + \frac{\lambda}{2}\Big(\|\mathbf{w}\|_2^2 + \|\boldsymbol{\theta}_{\overline{S}}\|_2^2\Big), \tag{5}$$

where $\mathbf{s}(\mathbf{w}) \circ \boldsymbol{\theta}$ denotes the Hadamard product, $\boldsymbol{\theta}_{\overline{S}} \in \mathbb{R}^{\overline{S}}$ is $\boldsymbol{\theta}$ restricted to indices in $\overline{S}$, and

$$\mathbf{s}_i(\mathbf{w}, \overline{S}) := \begin{cases} 1 & \text{if } i \in S, \\ \mathbf{w}_i & \text{if } i \notin S. \end{cases}$$

By a simple argument due to Hoff (2017), the objective function in (5) is equivalent to

$$\min_{\boldsymbol{\theta} \in \mathbb{R}^d} \|\mathbf{X}\boldsymbol{\theta} - \mathbf{y}\|_2^2 + \lambda \|\boldsymbol{\theta}_{\overline{S}}\|_1. \tag{6}$$

It follows that attention (or more generally overparameterization by trainable weights $\mathbf{w}$) can be seen as a way to implement $\ell_1$ regularization for least squares linear regression, i.e., the LASSO (Tibshirani, 1996). This connection between overparameterization and $\ell_1$ regularization has also been observed in several other recent works (Vaskevicius et al., 2019; Zhao et al., 2022; Tibshirani, 2021).

By this transformation and reasoning, regularized linear Sequential Attention can be seen as iteratively using the LASSO with $\ell_1$ regularization applied only to the unselected features—which is precisely the Sequential LASSO algorithm in Luo & Chen (2014). If we instead use $\mathrm{softmax}(\mathbf{w}, \overline{S})$ as in (1), then this only changes the choice of regularization, as shown in Lemma 3.2 (proof in Appendix A.3).

**Lemma 3.2.** *Let $D : \mathbb{R}^d \to \mathbb{R}^{\overline{S}}$ be the function defined by $D(\mathbf{w})_i = 1/\mathrm{softmax}_i^2(\mathbf{w}, \overline{S})$, for $i \in \overline{S}$. Denote its range and preimage by $\mathrm{ran}(D) \subseteq \mathbb{R}^{\overline{S}}$ and $D^{-1}(\cdot) \subseteq \mathbb{R}^d$, respectively. Moreover, define the functions $Q : \mathrm{ran}(D) \to \mathbb{R}$ and $Q^* : \mathbb{R}^{\overline{S}} \to \mathbb{R}$ by*

$$Q(\mathbf{q}) = \inf_{\mathbf{w} \in D^{-1}(\mathbf{q})} \|\mathbf{w}\|_2^2 \quad and \quad Q^*(\mathbf{x}) = \inf_{\mathbf{q} \in \mathrm{ran}(D)} \left( \sum_{i \in \overline{S}} \mathbf{x}_i \mathbf{q}_i + Q(\mathbf{q}) \right).$$

*Then, the following two optimization problems with respect to $\boldsymbol{\beta} \in \mathbb{R}^d$ are equivalent:*

$$\inf_{\substack{\boldsymbol{\beta} \in \mathbb{R}^d \\ \text{s.t. } \boldsymbol{\beta} = \mathrm{softmax}(\mathbf{w}, \overline{S}) \circ \boldsymbol{\theta} \\ \mathbf{w} \in \mathbb{R}^d, \boldsymbol{\theta} \in \mathbb{R}^d}} \|\mathbf{X}\boldsymbol{\beta} - \mathbf{y}\|_2^2 + \frac{\lambda}{2}\Big(\|\mathbf{w}\|_2^2 + \|\boldsymbol{\theta}_{\overline{S}}\|_2^2\Big) = \inf_{\boldsymbol{\beta} \in \mathbb{R}^d} \|\mathbf{X}\boldsymbol{\beta} - \mathbf{y}\|_2^2 + \frac{\lambda}{2}Q^*(\boldsymbol{\beta} \circ \boldsymbol{\beta}). \tag{7}$$

We present contour plots of $Q^*(\boldsymbol{\beta} \circ \boldsymbol{\beta})$ for $\boldsymbol{\beta} \in \mathbb{R}^2$ in Figure 2. These plots suggest that $Q^*(\boldsymbol{\beta} \circ \boldsymbol{\beta})$ is a concave regularizer when $|\beta_1| + |\beta_2| > 2$, which would thus approximate the $\ell_0$ regularizer and induce a sparse solution of $\boldsymbol{\beta}$ (Zhang & Zhang, 2012), as $\ell_1$ regularization does (Tibshirani, 1996).

## 3.2 Sequential LASSO and OMP

This connection between Sequential Attention and Sequential LASSO gives us a new perspective about how Sequential Attention works. The only known guarantee for Sequential LASSO, to the best of our knowledge, is a statistical recovery result when the input is a sparse linear combination with Gaussian noise in the ultra high-dimensional setting (Luo & Chen, 2014). This does not, however, fully explain why Sequential Attention is such an effective feature selection algorithm.

To bridge our main results, we prove a novel equivalence between Sequential LASSO and OMP.

**Theorem 3.3.** *Let* $\mathbf{X} \in \mathbb{R}^{n \times d}$ *be a design matrix with* $\ell_2$ *unit vector columns, and let* $\mathbf{y} \in \mathbb{R}^d$ *denote the response, also an* $\ell_2$ *unit vector. The Sequential LASSO algorithm maintains a set of features* $S \subseteq [d]$ *such that, at each feature selection step, it selects a feature* $i \in \overline{S}$ *such that*

$$\left| \langle \mathbf{X}_i, \mathbf{P}_S^\perp \mathbf{y} \rangle \right| = \left\| \mathbf{X}^\top \mathbf{P}_S^\perp \mathbf{y} \right\|_\infty,$$

*where* $\mathbf{X}_S$ *is the* $n \times |S|$ *matrix given formed by the columns of* $\mathbf{X}$ *indexed by* $S$, *and* $\mathbf{P}_S^\perp$ *is the projection matrix onto the orthogonal complement of the span of* $\mathbf{X}_S$.

Note that this is extremely close to saying that Sequential LASSO and OMP select the exact same set of features. The only difference appears when there are multiple features with norm $\|\mathbf{X}^\top \mathbf{P}_S^\perp \mathbf{y}\|_\infty$. In this case, it is possible that Sequential LASSO chooses the next feature from a set of features that is strictly smaller than the set of features from which OMP chooses, so the "tie-breaking" can differ between the two algorithms. In practice, however, this rarely happens. For instance, if only one feature is selected at each step, which is the case with probability 1 if random continuous noise is added to the data, then Sequential LASSO and OMP will select the exact same set of features.

**Remark 3.4.** It was shown in (Luo & Chen, 2014) that Sequential LASSO is equivalent to OMP in the statistical recovery regime, i.e., when $\mathbf{y} = \mathbf{X}\boldsymbol{\beta}^* + \boldsymbol{\varepsilon}$ for some true sparse weight vector $\boldsymbol{\beta}^*$ and i.i.d. Gaussian noise $\boldsymbol{\varepsilon} \sim \mathcal{N}(0, \sigma\mathbf{I}_n)$, under an ultra high-dimensional regime where the dimension $d$ is exponential in the number of examples $n$. We prove this equivalence in the *fully general setting*.

The argument below shows that Sequential LASSO and OMP are equivalent, thus establishing that regularized linear Sequential Attention and Sequential LASSO have the same approximation guarantees as OMP.

**Geometry of Sequential LASSO.** We first study the geometry of optimal solutions to Equation (4). Let $S \subseteq [d]$ be the set of currently selected features. Following work on the LASSO in Tibshirani & Taylor (2011), we rewrite (4) as the following constrained optimization problem:

$$\min_{\mathbf{z} \in \mathbb{R}^n, \boldsymbol{\beta} \in \mathbb{R}^d} \quad \frac{1}{2}\|\mathbf{z} - \mathbf{y}\|_2^2 + \lambda\|\boldsymbol{\beta}_{\overline{S}}\|_1 \tag{8}$$
$$\text{subject to} \quad \mathbf{z} = \mathbf{X}\boldsymbol{\beta}.$$

It can then be shown that the dual problem is equivalent to finding the projection, i.e., closest point in Euclidean distance, $\mathbf{u} \in \mathbb{R}^n$ of $\mathbf{P}_S^\perp \mathbf{y}$ onto the polyhedral section $C_\lambda \cap \text{colspan}(\mathbf{X}_S)^\perp$, where

$$C_\lambda := \left\{ \mathbf{u}' \in \mathbb{R}^n : \left\| \mathbf{X}^\top \mathbf{u}' \right\|_\infty \le \lambda \right\}$$

and $\text{colspan}(\mathbf{X}_S)^\perp$ denotes the orthogonal complement of $\text{colspan}(\mathbf{X}_S)$. See Appendix A.1 for the full details. The primal and dual variables are related through $\mathbf{z}$ by

$$\mathbf{X}\boldsymbol{\beta} = \mathbf{z} = \mathbf{y} - \mathbf{u}. \tag{9}$$

**Selection of features in Sequential LASSO.** Next, we analyze how Sequential LASSO selects its features. Let $\boldsymbol{\beta}_S^* = \mathbf{X}_S^+ \mathbf{y}$ be the optimal solution for features restricted in $S$. Then, subtracting $\mathbf{X}_S\boldsymbol{\beta}_S^*$ from both sides of (9) gives

$$\mathbf{X}\boldsymbol{\beta} - \mathbf{X}_S\boldsymbol{\beta}_S^* = \mathbf{y} - \mathbf{X}_S\boldsymbol{\beta}_S^* - \mathbf{u}$$
$$= \mathbf{P}_S^\perp \mathbf{y} - \mathbf{u}. \tag{10}$$

Note that if $\lambda \ge \|\mathbf{X}^\top \mathbf{P}_S^\perp \mathbf{y}\|_\infty$, then the projection of $\mathbf{P}_S^\perp \mathbf{y}$ onto $C_\lambda$ is just $\mathbf{u} = \mathbf{P}_S^\perp \mathbf{y}$, so by (10),

$$\mathbf{X}\boldsymbol{\beta} - \mathbf{X}_S\boldsymbol{\beta}_S^* = \mathbf{P}_S^\perp \mathbf{y} - \mathbf{P}_S^\perp \mathbf{y} = \mathbf{0},$$

meaning that $\boldsymbol{\beta}$ is zero outside of $S$. We now show that for $\lambda$ slightly smaller than $\|\mathbf{X}^\top \mathbf{P}_S^\perp \mathbf{y}\|_\infty$, the residual $\mathbf{P}_S^\perp \mathbf{y} - \mathbf{u}$ is in the span of features $\mathbf{X}_i$ that maximize the correlation with $\mathbf{P}_S^\perp \mathbf{y}$.

**Lemma 3.5** (Projection residuals of the Sequential LASSO). *Let* $\mathbf{p}$ *denote the projection of* $\mathbf{P}_S^\perp \mathbf{y}$ *onto* $C_\lambda \cap \text{colspan}(\mathbf{X}_S)^\perp$. *There exists* $\lambda_0 < \left\| \mathbf{X}^\top \mathbf{P}_S^\perp \mathbf{y} \right\|_\infty$ *such that for all* $\lambda \in (\lambda_0, \|\mathbf{X}^\top \mathbf{P}_S^\perp \mathbf{y}\|_\infty)$ *the residual* $\mathbf{P}_S^\perp \mathbf{y} - \mathbf{p}$ *lies on* $\text{colspan}(\mathbf{X}_T)$, *for*

$$T := \left\{ i \in [d] : \left| \langle \mathbf{X}_i, \mathbf{P}_S^\perp \mathbf{y} \rangle \right| = \left\| \mathbf{X}^\top \mathbf{P}_S^\perp \mathbf{y} \right\|_\infty \right\}.$$

We defer the proof of Lemma 3.5 to Appendix A.2.

By Lemma 3.5 and (10), the optimal $\boldsymbol{\beta}$ when selecting the next feature has the following properties:

1. if $i \in S$, then $\boldsymbol{\beta}_i$ is equal to the $i$-th value in the previous solution $\boldsymbol{\beta}_S^*$; and
2. if $i \notin S$, then $\boldsymbol{\beta}_i$ can be nonzero only if $i \in T$.

It follows that Sequential LASSO selects a feature that maximizes the correlation $|\langle \mathbf{X}_j, \mathbf{P}_S^\perp \mathbf{y} \rangle|$, just as OMP does. Thus, we have shown an equivalence between Sequential LASSO and OMP without any additional assumptions.

## 4 EXPERIMENTS

### 4.1 FEATURE SELECTION FOR NEURAL NETWORKS

**Small-scale experiments.** We investigate the performance of Sequential Attention, as presented in Algorithm 1, through experiments on standard feature selection benchmarks for neural networks. In these experiments, we consider six datasets used in experiments in Lemhadri et al. (2021); Balın et al. (2019), and select $k = 50$ features using a one-layer neural network with hidden width 67 and ReLU activation (just as in these previous works). For more points of comparison, we also implement the attention-based feature selection algorithms of Balın et al. (2019); Liao et al. (2021) and the Group LASSO, which has been considered in many works that aim to sparsifiy neural networks as discussed in Section 1.1. We also implement natural adaptations of the Sequential LASSO and OMP for neural networks and evaluate their performance.

In Figure 3, we see that Sequential Attention is competitive with or outperforms all feature selection algorithms on this benchmark suite. For each algorithm, we report the mean of the prediction accuracies averaged over five feature selection trials. We provide more details about the experimental setup in Appendix B.2, including specifications about each dataset in Table 1 and the mean prediction accuracies with their standard deviations in Table 2. We also visualize the selected features on MNIST (i.e., pixels) in Figure 5.

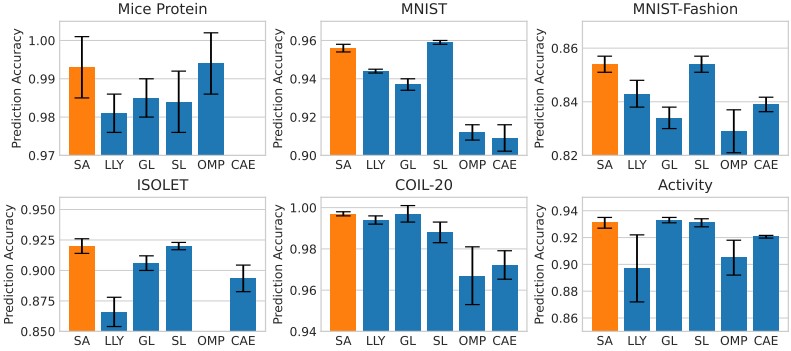

Figure 3: Feature selection results for small-scale neural network experiments. Here, SA = Sequential Attention, LLY = (Liao et al., 2021), GL = Group LASSO, SL = Sequential LASSO, OMP = OMP, and CAE = Concrete Autoencoder (Balın et al., 2019).

We note that our algorithm is considerably more efficient compared to prior feature selection algorithms, especially those designed for neural networks. This is because many of these prior algorithms introduce entire subnetworks to train (Balın et al., 2019; Gui et al., 2019; Wojtas & Chen, 2020; Liao et al., 2021), whereas Sequential Attention only adds $d$ additional trainable variables. Furthermore, in these experiments, we implement an optimized version of Algorithm 1 that only trains one model rather than $k$ models, by partitioning the training epochs into $k$ parts and selecting one feature in each of these $k$ parts. Combining these two aspects makes for an extremely efficient algorithm. We provide an evaluation of the running time efficiency of Sequential Attention in Appendix B.2.3.

**Large-scale experiments.** To demonstrate the scalability of our algorithm, we perform large-scale feature selection experiments on the Criteo click dataset, which consists of 39 features and over three

billion examples for predicting click-through rates (Diemert Eustache, Meynet Julien et al., 2017). Our results in Figure 4 show that Sequential Attention outperforms other methods when at least 15 features are selected. In particular, these plots highlight the fact that Sequential Attention excels at finding valuable features once a few features are already in the model, and that it has substantially less variance than LASSO-based feature selection algorithms. See Appendix B.3 for further discussion.

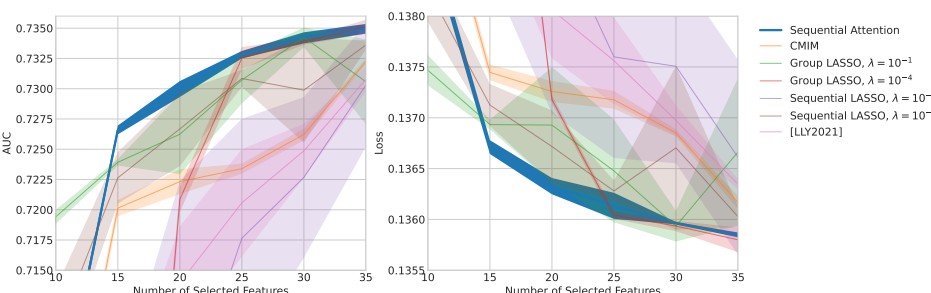

Figure 4: AUC and log loss when selecting $k \in \{10, 15, 20, 25, 30, 35\}$ features for Criteo dataset.

## 4.2 THE ROLE OF HADAMARD PRODUCT OVERPARAMETERIZATION IN ATTENTION

In Section 1, we argued that Sequential Attention has provable guarantees for least squares linear regression by showing that a version that removes the softmax and introduces $\ell_2$ regularization results in an algorithm that is equivalent to OMP. Thus, there is a gap between the implementation of Sequential Attention in Algorithm 1 and our theoretical analysis. We empirically bridge this gap by showing that regularized linear Sequential Attention yields results that are almost indistinguishable to the original version. In Figure 10 (Appendix B.5), we compare the following Hadamard product overparameterization schemes:

- softmax: as described in Section 1
- $\ell_1$: $\mathbf{s}_i(\mathbf{w}) = |\mathbf{w}_i|$ for $i \in \overline{S}$, which captures the provable variant discussed in Section 1
- $\ell_2$: $\mathbf{s}_i(\mathbf{w}) = |\mathbf{w}_i|^2$ for $i \in \overline{S}$
- $\ell_1$ normalized: $\mathbf{s}_i(\mathbf{w}) = |\mathbf{w}_i| / \sum_{j \in \overline{S}} |\mathbf{w}_j|$ for $i \in \overline{S}$
- $\ell_2$ normalized: $\mathbf{s}_i(\mathbf{w}) = |\mathbf{w}_i|^2 / \sum_{j \in \overline{S}} |\mathbf{w}_j|^2$ for $i \in \overline{S}$

Further, for each of the benchmark datasets, all of these variants outperform LassoNet and the other baselines considered in Lemhadri et al. (2021). See Appendix B.5 for more details.

## 5 CONCLUSION

This work introduces Sequential Attention, an adaptive attention-based feature selection algorithm designed in part for neural networks. Empirically, Sequential Attention improves significantly upon previous methods on widely-used benchmarks. Theoretically, we show that a relaxed variant of Sequential Attention is equivalent to Sequential LASSO (Luo & Chen, 2014). In turn, we prove a novel connection between Sequential LASSO and Orthogonal Matching Pursuit, thus transferring the provable guarantees of OMP to Sequential Attention and shedding light on our empirical results. This analysis also provides new insights into the the role of attention for feature selection via adaptivity, overparameterization, and connections to marginal gains.

We conclude with a number of open questions that stem from this work. The first question concerns the generalization of our theoretical results for Sequential LASSO to other models. OMP admits provable guarantees for a wide class of generalized linear models (Elenberg et al., 2018), so is the same true for Sequential LASSO? Our second question concerns the role of softmax in Algorithm 1. Our experimental results suggest that using softmax for overparametrization may not be necessary, and that a wide variety of alternative expressions can be used. On the other hand, our provable guarantees only hold for the overparameterization scheme in the regularized linear Sequential Attention algorithm (see Definition 3.1). Can we obtain a deeper understanding about the pros and cons of the softmax and other overparameterization patterns, both theoretically and empirically?

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

# A  MISSING PROOFS FROM SECTION 3

## A.1  LAGRANGIAN DUAL OF SEQUENTIAL LASSO

We first show that the Lagrangian dual of (8) is equivalent to the following problem:

$$
\begin{aligned}
\min_{\mathbf{u}\in\mathbb{R}^n} \quad & \frac{1}{2}\|\mathbf{y}-\mathbf{u}\|_2^2 \\
\text{subject to} \quad & \left\|\mathbf{X}^\top\mathbf{u}\right\|_\infty \leq \lambda \\
& \mathbf{X}_j^\top\mathbf{u} = 0, \qquad \forall j \in S
\end{aligned}
\tag{11}
$$

We then use the Pythagorean theorem to replace $\mathbf{y}$ by $\mathbf{P}_S^\perp\mathbf{y}$.

First consider the Lagrangian dual problem:

$$
\max_{\mathbf{u}\in\mathbb{R}^n}\min_{\mathbf{z}\in\mathbb{R}^n,\boldsymbol{\beta}\in\mathbb{R}^d} \frac{1}{2}\|\mathbf{z}-\mathbf{y}\|_2^2 + \lambda\|\boldsymbol{\beta}|_{\overline{S}}\|_1 + \mathbf{u}^\top(\mathbf{z}-\mathbf{X}\boldsymbol{\beta}).
\tag{12}
$$

Note that the primal problem is strictly feasible and convex, so strong duality holds (see, e.g., Section 5.2.3 of Boyd & Vandenberghe (2004)). Considering just the terms involving the variable $\mathbf{z}$ in (12), we have that

$$
\begin{aligned}
\frac{1}{2}\|\mathbf{z}-\mathbf{y}\|_2^2 + \mathbf{u}^\top\mathbf{z} &= \frac{1}{2}\|\mathbf{z}\|_2^2 - (\mathbf{y}-\mathbf{u})^\top\mathbf{z} + \frac{1}{2}\|\mathbf{y}\|_2^2 \\
&= \frac{1}{2}\|\mathbf{z}-(\mathbf{y}-\mathbf{u})\|_2^2 + \frac{1}{2}\|\mathbf{y}\|_2^2 - \frac{1}{2}\|\mathbf{y}-\mathbf{u}\|_2^2,
\end{aligned}
$$

which is minimized at $\mathbf{z}=\mathbf{y}-\mathbf{u}$ as $\mathbf{z}$ varies over $\mathbb{R}^n$. On the other hand, consider just the terms involving the variable $\boldsymbol{\beta}$ in (12), that is,

$$
\lambda\|\boldsymbol{\beta}_{\overline{S}}\|_1 - \mathbf{u}^\top\mathbf{X}\boldsymbol{\beta}.
\tag{13}
$$

Note that if $\mathbf{X}^\top\mathbf{u}$ is nonzero on any coordinate in $S$, then (13) can be made arbitrarily negative by setting $\boldsymbol{\beta}_{\overline{S}}$ to be zero and $\boldsymbol{\beta}_S$ appropriately. Similarly, if $\|\mathbf{X}^\top\mathbf{u}\|_\infty > \lambda$, then (13) can also be made to be arbitrarily negative. On the other hand, if $(\mathbf{X}^\top\mathbf{u})_S = \mathbf{0}$ and $\left\|\mathbf{X}^\top\mathbf{u}\right\|_\infty \leq \lambda$, then (13) is minimized at 0. This gives the dual in Equation (11).

We now show that by the Pythagorean theorem, we can project $\mathbf{P}_S^\perp\mathbf{y}$ in (11) rather than $\mathbf{y}$. In (11), recall that $\mathbf{u}$ is constrained to be in $\mathrm{colspan}(\mathbf{X}_S)^\perp$. Then, by the Pythagorean theorem, we have

$$
\begin{aligned}
\frac{1}{2}\|\mathbf{y}-\mathbf{u}\|_2^2 &= \frac{1}{2}\left\|\mathbf{y}-\mathbf{P}_S^\perp\mathbf{y}+\mathbf{P}_S^\perp\mathbf{y}-\mathbf{u}\right\|_2^2 \\
&= \frac{1}{2}\left\|\mathbf{y}-\mathbf{P}_S^\perp\mathbf{y}\right\|_2^2 + \frac{1}{2}\left\|\mathbf{P}_S^\perp\mathbf{y}-\mathbf{u}\right\|_2^2,
\end{aligned}
$$

since $\mathbf{y}-\mathbf{P}_S^\perp\mathbf{y} = \mathbf{P}_S\mathbf{y}$ is orthogonal to $\mathrm{colspan}(\mathbf{X}_S)^\perp$ and both $\mathbf{P}_S^\perp\mathbf{y}$ and $\mathbf{u}$ are in $\mathrm{colspan}(\mathbf{X}_S)^\perp$. The first term in the above does not depend on $\mathbf{u}$ and thus we may discard it. Our problem therefore reduces to projecting $\mathbf{P}_S^\perp\mathbf{y}$ onto $C_\lambda \cap \mathrm{colspan}(\mathbf{X}_S)^\perp$, rather than $\mathbf{y}$.

## A.2  PROOF OF LEMMA 3.5

*Proof of Lemma 3.5.* Our approach is to reduce the projection of $\mathbf{P}_S^\perp\mathbf{y}$ onto the polytope defined by $C_\lambda \cap \mathrm{colspan}(\mathbf{X})^\perp$ to a projection onto an affine space.

We first argue that it suffices to project onto the faces of $C_\lambda$ specified by set $T$. For $\lambda > 0$, feature indices $i \in [d]$, and signs $\pm$, we define the faces

$$
F_{\lambda,i,\pm} := \{\mathbf{u}\in\mathbb{R}^n : \pm\langle\mathbf{X}_i,\mathbf{u}\rangle = \lambda\}
$$

of $C_\lambda$. Let $\lambda = (1-\varepsilon)\|\mathbf{X}^\top\mathbf{P}_S^\perp\mathbf{y}\|_\infty$, for $\varepsilon > 0$ to be chosen sufficiently small. Then clearly

$$
(1-\varepsilon)\mathbf{P}_S^\perp\mathbf{y} \in C_\lambda \cap \mathrm{colspan}(\mathbf{X}_S)^\perp,
$$

so

$$\min_{\mathbf{u} \in C_\lambda \cap \mathrm{colspan}(\mathbf{X}_S)^\perp} \left\|\mathbf{P}_S^\perp \mathbf{y} - \mathbf{u}\right\|_2^2 \leq \left\|\mathbf{P}_S^\perp \mathbf{y} - (1 - \varepsilon)\mathbf{P}_S^\perp \mathbf{y}\right\|_2^2$$

$$= \varepsilon^2 \left\|\mathbf{P}_S^\perp \mathbf{y}\right\|_2^2.$$

In fact, $(1 - \varepsilon)\mathbf{P}_S^\perp \mathbf{y}$ lies on the intersection of faces $F_{\lambda, i, \pm}$ for an appropriate choice of signs and $i \in T$. Without loss of generality, we assume that these faces are just $F_{\lambda, i, +}$ for $i \in T$. Note also that for any $i \notin T$,

$$\min_{\mathbf{u} \in F_{\lambda, i, \pm}} \left\|\mathbf{P}_S^\perp \mathbf{y} - \mathbf{u}\right\|_2^2 \geq \min_{\mathbf{u} \in F_{\lambda, i, \pm}} \left\langle \mathbf{X}_i, \mathbf{P}_S^\perp \mathbf{y} - \mathbf{u}\right\rangle^2 \qquad \text{(Cauchy–Schwarz, } \|\mathbf{X}_i\|_2 \leq 1)$$

$$= \min_{\mathbf{u} \in F_{\lambda, i, \pm}} \left|\mathbf{X}_i^\top \mathbf{P}_S^\perp \mathbf{y} - \mathbf{X}_i^\top \mathbf{u}\right|^2$$

$$= \left(\left|\mathbf{X}_i^\top \mathbf{P}_S^\perp \mathbf{y}\right| - \lambda\right)^2 \qquad (\mathbf{u} \in F_{\lambda, i, \pm})$$

$$\geq \left((1 - \varepsilon)\left\|\mathbf{X}^\top \mathbf{P}_S^\perp \mathbf{y}\right\|_\infty - \left\|\mathbf{X}_T^\top \mathbf{P}_S^\perp \mathbf{y}\right\|_\infty\right)^2.$$

For all $\varepsilon < \varepsilon_0$, for $\varepsilon_0$ small enough, this is larger than $\varepsilon^2 \|\mathbf{P}_S^\perp \mathbf{y}\|_2^2$. Thus, for $\varepsilon$ small enough, $\mathbf{P}_S^\perp \mathbf{y}$ is closer to the faces $F_{\lambda, i, +}$ for $i \in T$ than any other face. Therefore, we set $\lambda_0 = (1 - \varepsilon_0)\|\mathbf{X}^\top \mathbf{P}_S^\perp \mathbf{y}\|_\infty$.

Now, by the complementary slackness of the KKT conditions for the projection $\mathbf{u}$ of $\mathbf{P}_S^\perp \mathbf{y}$ onto $C_\lambda$, for each face of $C_\lambda$ we either have that $\mathbf{u}$ lies on the face or that the projection does not change if we remove the face. For $i \notin T$, note that by the above calculation, the projection $\mathbf{u}$ cannot lie on $F_{\lambda, i, \pm}$, so $\mathbf{u}$ is simply the projection onto

$$C' = \left\{\mathbf{u} \in \mathbb{R}^n : \mathbf{X}_T^\top \mathbf{u} \leq \lambda \mathbf{1}_T\right\}.$$

By reversing the dual problem reasoning from before, the residual of the projection onto $C'$ must lie on the column span of $\mathbf{X}_T$. □

### A.3 Parameterization patterns and regularization

*Proof of Lemma 3.2.* The optimization problem on the left-hand side of Equation (7) with respect to $\boldsymbol{\beta}$ is equivalent to

$$\inf_{\boldsymbol{\beta} \in \mathbb{R}^d} \left(\|\mathbf{X}\boldsymbol{\beta} - \mathbf{y}\|_2^2 + \frac{\lambda}{2} \inf_{\mathbf{w} \in \mathbb{R}^d} \left(\|\mathbf{w}\|_2^2 + \sum_{i \in \overline{S}} \frac{\boldsymbol{\beta}_i^2}{\mathbf{s}_i(\mathbf{w})^2}\right)\right). \tag{14}$$

If we define

$$\tilde{Q}^*(\mathbf{x}) = \inf_{\mathbf{w} \in \mathbb{R}^d} \left(\|\mathbf{w}\|_2^2 + \sum_{i \in \overline{S}} \frac{\mathbf{x}_i}{\mathbf{s}_i(\mathbf{w})^2}\right),$$

then the LHS of (7) and (14) are equivalent to $\inf_{\boldsymbol{\beta} \in \mathbb{R}^d} (\|\mathbf{X}\boldsymbol{\beta} - \mathbf{y}\|_2^2 + \frac{\lambda}{2}\tilde{Q}^*(\boldsymbol{\beta} \circ \boldsymbol{\beta}))$. Re-parameterizing the minimization problem in the definition of $\tilde{Q}^*(\mathbf{x})$ (by setting $\mathbf{q} = D(\mathbf{w})$), we obtain $\tilde{Q}^* = Q^*$. □

## B Additional experiments

### B.1 Visualization of selected MNIST features

In Figure 5, we present visualizations of the features (i.e., pixels) selected by Sequential Attention and the baseline algorithms. This provides some intuition on the nature of the features that these algorithms select. Similar visualizations for MNIST can be found in works such as Balın et al. (2019); Gui et al. (2019); Wojtas & Chen (2020); Lemhadri et al. (2021); Liao et al. (2021). Note that these visualizations serve as a basic sanity check about the kinds of pixels that these algorithms select. For instance, the degree to which the selected pixels are "clustered" can be used to informally assess the redundancy of features selected for image datasets, since neighboring pixels tend to represent

redundant information. It is also useful at time to assess which regions of the image are selected. For example, the central regions of the MNIST images are more informative than the edges.

Sequential Attention selects a highly diverse set of pixels due to its adaptivity. Sequential LASSO also selects a very similar set of pixels, as suggested by our theoretical analysis in Section 3. Curiously, OMP does not yield a competitive set of pixels, which demonstrates that OMP does not generalize well from least squares regression and generalized linear models to deep neural networks.

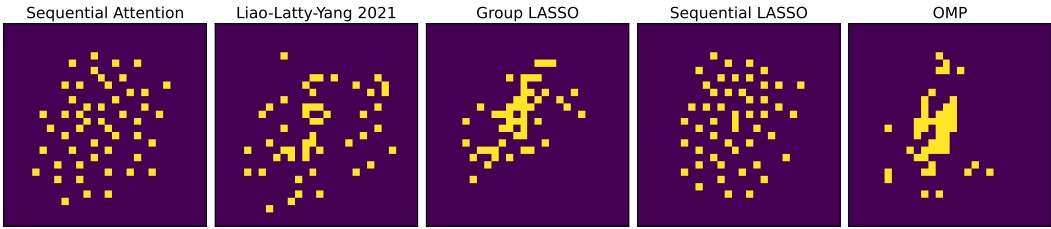

Figure 5: Visualizations of the $k = 50$ pixels selected by the feature selection algorithms on MNIST.

## B.2 ADDITIONAL DETAILS ON SMALL-SCALE EXPERIMENTS

We start by presenting details about each of the datasets used for neural network feature selection in Balın et al. (2019) and Lemhadri et al. (2021) in Table 1.

Table 1: Statistics about benchmark datasets.

| Dataset | # Examples | # Features | # Classes | Type |
|---|---|---|---|---|
| Mice Protein | 1,080 | 77 | 8 | Biology |
| MNIST | 60,000 | 784 | 10 | Image |
| MNIST-Fashion | 60,000 | 784 | 10 | Image |
| ISOLET | 7,797 | 617 | 26 | Speech |
| COIL-20 | 1,440 | 400 | 20 | Image |
| Activity | 5,744 | 561 | 6 | Sensor |

In Figure 3, the error bars are computed using the standard deviation over five runs of the algorithm with different random seeds. The values used to generate these plots are provided below in Table 2.

Table 2: Feature selection results for small-scale datasets (see Figure 3 for a key). These values are the average prediction accuracies on the test data and their standard deviations.

| Dataset | SA | LLY | GL | SL | OMP | CAE |
|---|---|---|---|---|---|---|
| Mice Protein | $0.993 \pm 0.008$ | $0.981 \pm 0.005$ | $0.985 \pm 0.005$ | $0.984 \pm 0.008$ | $0.994 \pm 0.008$ | $0.956 \pm 0.012$ |
| MNIST | $0.956 \pm 0.002$ | $0.944 \pm 0.001$ | $0.937 \pm 0.003$ | $0.959 \pm 0.001$ | $0.912 \pm 0.004$ | $0.909 \pm 0.007$ |
| MNIST-Fashion | $0.854 \pm 0.003$ | $0.843 \pm 0.005$ | $0.834 \pm 0.004$ | $0.854 \pm 0.003$ | $0.829 \pm 0.008$ | $0.839 \pm 0.003$ |
| ISOLET | $0.920 \pm 0.006$ | $0.866 \pm 0.012$ | $0.906 \pm 0.006$ | $0.920 \pm 0.003$ | $0.727 \pm 0.026$ | $0.893 \pm 0.011$ |
| COIL-20 | $0.997 \pm 0.001$ | $0.994 \pm 0.002$ | $0.997 \pm 0.004$ | $0.988 \pm 0.005$ | $0.967 \pm 0.014$ | $0.972 \pm 0.007$ |
| Activity | $0.931 \pm 0.004$ | $0.897 \pm 0.025$ | $0.933 \pm 0.002$ | $0.931 \pm 0.003$ | $0.905 \pm 0.013$ | $0.921 \pm 0.001$ |

### B.2.1 MODEL ACCURACIES WITH ALL FEATURES

To adjust for the differences between the values reported in Lemhadri et al. (2021) and ours due (e.g., due to factors such as the implementation framework), we list the accuracies obtained by training the models with all of the available features in Table 3.

### B.2.2 GENERALIZING OMP TO NEURAL NETWORKS

As stated in Algorithm 2, it may be difficult to see exactly how OMP generalizes from a linear regression model to neural networks. To do this, first observe that OMP naturally generalizes to

Table 3: Model accuracies when trained using all available features.

| Dataset | Lemhadri et al. (2021) | This paper |
|---|---|---|
| Mice Protein | 0.990 | 0.963 |
| MNIST | 0.928 | 0.953 |
| MNIST-Fashion | 0.833 | 0.869 |
| ISOLET | 0.953 | 0.961 |
| COIL-20 | 0.996 | 0.986 |
| Activity | 0.853 | 0.954 |

generalized linear models (GLMs) via the gradient of the link function, as shown in Elenberg et al. (2018). Then, to extend this to neural networks, we view the neural network as a GLM for any fixing of the hidden layer weights, and then we use the gradient of this GLM with respect to the inputs as the feature importance scores.

### B.2.3 EFFICIENCY EVALUATION

In this subsection, we evaluate the efficiency of the Sequential Attention algorithm against our other baseline algorithms. We do so by fixing the number of epochs and batch size for all of the algorithms, and then evaluating the accuracy as well as the wall clock time of each algorithm. Figures 6 and 7 provide a visualization of the accuracy and wall clock time of feature selection, while Tables 5 and 6 provide the average and standard deviations. Table 4 provides the epochs and batch size settings that were fixed for these experiments.

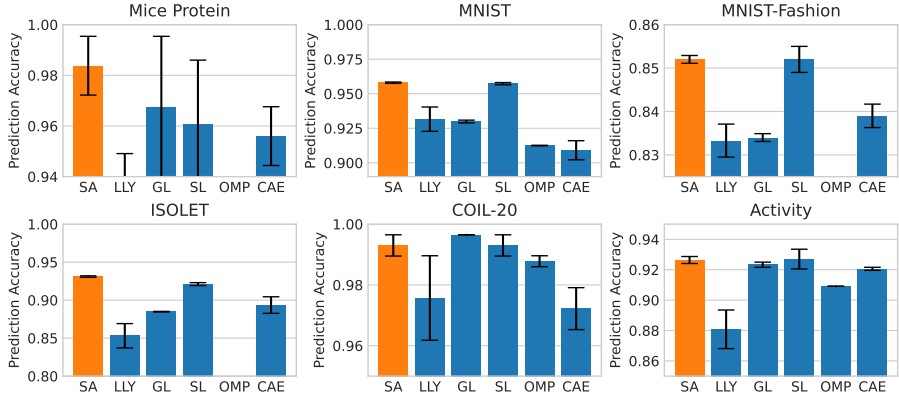

Figure 6: Feature selection accuracy for efficiency evaluation.

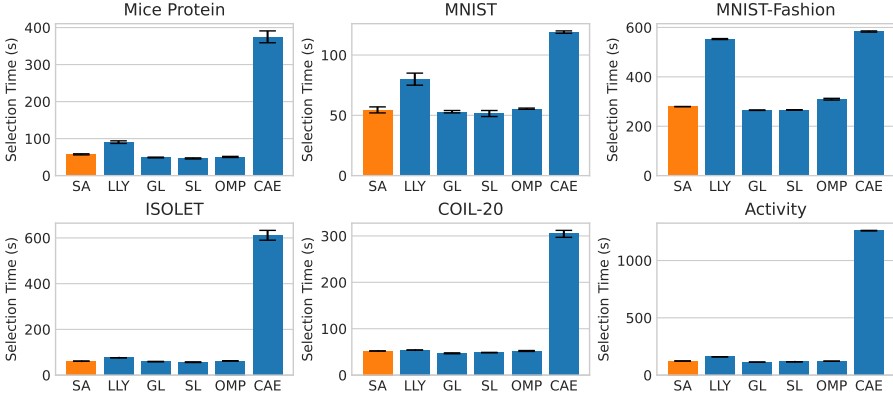

Figure 7: Feature selection wall clock time in seconds for efficiency evaluation.

Table 4: Epochs and batch size used to compare the efficiency of feature selection algorithms.

| Dataset | Epochs | Batch Size |
|---|---|---|
| Mice Protein | 2000 | 256 |
| MNIST | 50 | 256 |
| MNIST-Fashion | 250 | 128 |
| ISOLET | 500 | 256 |
| COIL-20 | 1000 | 256 |
| Activity | 1000 | 512 |

Table 5: Feature selection accuracy for efficiency evaluation. We report the mean accuracy on the test dataset and the the standard deviation across five trials.

| Dataset | SA | LLY | GL | SL | OMP | CAE |
|---|---|---|---|---|---|---|
| Mice Protein | $\mathbf{0.984 \pm 0.012}$ | $0.907 \pm 0.042$ | $0.968 \pm 0.028$ | $0.961 \pm 0.025$ | $0.556 \pm 0.032$ | $0.956 \pm 0.012$ |
| MNIST | $\mathbf{0.958 \pm 0.001}$ | $0.932 \pm 0.009$ | $0.930 \pm 0.001$ | $\mathbf{0.957 \pm 0.001}$ | $0.912 \pm 0.000$ | $0.909 \pm 0.007$ |
| MNIST-Fashion | $\mathbf{0.852 \pm 0.001}$ | $0.833 \pm 0.004$ | $0.834 \pm 0.001$ | $\mathbf{0.852 \pm 0.003}$ | $0.722 \pm 0.029$ | $0.839 \pm 0.003$ |
| ISOLET | $\mathbf{0.931 \pm 0.001}$ | $0.853 \pm 0.016$ | $0.885 \pm 0.000$ | $0.921 \pm 0.002$ | $0.580 \pm 0.025$ | $0.893 \pm 0.011$ |
| COIL-20 | $\mathbf{0.993 \pm 0.004}$ | $0.976 \pm 0.014$ | $0.997 \pm 0.000$ | $\mathbf{0.993 \pm 0.004}$ | $0.988 \pm 0.002$ | $0.972 \pm 0.007$ |
| Activity | $\mathbf{0.926 \pm 0.002}$ | $0.881 \pm 0.013$ | $0.923 \pm 0.002$ | $\mathbf{0.927 \pm 0.006}$ | $0.909 \pm 0.000$ | $0.921 \pm 0.001$ |

Table 6: Feature selection wall clock time in seconds for efficiency evaluations. These values are the mean wall clock time on the test dataset and their standard deviation across five trials.

| Dataset | SA | LLY | GL | SL | OMP | CAE |
|---|---|---|---|---|---|---|
| Mice Protein | $57.5 \pm 1.5$ | $90.5 \pm 3.5$ | $49.0 \pm 1.0$ | $46.5 \pm 1.5$ | $50.5 \pm 1.5$ | $375.0 \pm 16.0$ |
| MNIST | $54.5 \pm 2.5$ | $80.0 \pm 5.0$ | $53.0 \pm 1.0$ | $51.5 \pm 2.5$ | $55.5 \pm 0.5$ | $119.0 \pm 1.0$ |
| MNIST-Fashion | $279.5 \pm 0.5$ | $553.0 \pm 2.0$ | $265.0 \pm 1.0$ | $266.0 \pm 1.0$ | $309.5 \pm 3.5$ | $583.5 \pm 2.5$ |
| ISOLET | $61.0 \pm 0.0$ | $76.0 \pm 0.0$ | $59.0 \pm 1.0$ | $56.5 \pm 1.5$ | $62.0 \pm 1.0$ | $611.5 \pm 21.5$ |
| COIL-20 | $52.0 \pm 0.0$ | $54.0 \pm 0.0$ | $47.0 \pm 1.0$ | $48.5 \pm 0.5$ | $52.0 \pm 1.0$ | $304.5 \pm 7.5$ |
| Activity | $123.0 \pm 1.0$ | $159.5 \pm 0.5$ | $113.5 \pm 0.5$ | $116.0 \pm 0.0$ | $121.5 \pm 0.5$ | $1260.5 \pm 2.5$ |

### B.2.4    NOTES ON THE ONE-PASS IMPLEMENTATION

We make several remarks about the one-pass implementation of Sequential Attention. First, as noted in Section 4.1, our practical implementation of Sequential Attention only trains one model instead of $k$ models. We do this by partitioning the training epochs into $k$ parts and selecting one part in each phase. This clearly gives a more efficient running time than training $k$ separate models. Similarly, we allow for a "warm-up" period prior to the feature selection phase, in which a small fraction of the training epochs are allotted to training just the neural network weights. When we do this one-pass implementation, we observe that it is important to reset the attention weights after each of the sequential feature selection phases, but resetting the neural network weights is not crucial for good performance.

Second, we note that when there is a large number of candidate features $d$, the softmax mask severely scales down the gradient updates to the model weights, which can lead to inefficient training. In these cases, it becomes important to prevent this by either using a temperature parameter in the softmax to counteract the small softmax values or by adjusting the learning rate to be high enough. Note that these two approaches can be considered to be equivalent.

### B.3    LARGE-SCALE EXPERIMENTS

In this section, we provide more details and discussion on our Criteo large dataset results. For this experiment, we use a dense neural network with 768, 256, and 128 neurons in each of the three hidden layers with ReLU activations. In Figure 4, the error bars are generated as the standard deviation over running the algorithm three times with different random seeds, and the shadowed regions linearly interpolate between these error bars. The values used to generate the plot are provided in Table 7 and Table 8.

We first note that this dataset is so large that it is expensive to make multiple passes through the dataset. Therefore, we modify the algorithms (both Sequential Attention and the other baselines) to make only one pass through the data by using disjoint fractions of the data for different "steps" of the algorithm. Hence, we select $k$ features while only "training" one model.

Table 7: AUC of Criteo large experiments. SA is Sequential Attention, GL is generalized LASSO, and SL is Sequential LASSO. The values in the header for the LASSO methods are the $\ell_1$ regularization strengths used for each method.

| $k$ | SA | CMIM | GL ($\lambda = 10^{-1}$) | GL ($\lambda = 10^{-4}$) | SL ($\lambda = 10^{-1}$) | SL ($\lambda = 10^{-4}$) | Liao et al. (2021) |
|---|---|---|---|---|---|---|---|
| 5 | $0.67232 \pm 0.00015$ | $0.63950 \pm 0.00076$ | $0.68342 \pm 0.00585$ | $0.50161 \pm 0.00227$ | $0.60278 \pm 0.04473$ | $0.67710 \pm 0.00873$ | $0.58300 \pm 0.06360$ |
| 10 | $0.70167 \pm 0.00060$ | $0.69402 \pm 0.00052$ | $0.71942 \pm 0.00059$ | $0.64262 \pm 0.00187$ | $0.62263 \pm 0.06097$ | $0.70964 \pm 0.00385$ | $0.68103 \pm 0.00137$ |
| 15 | $0.72659 \pm 0.00036$ | $0.72014 \pm 0.00067$ | $0.72392 \pm 0.00027$ | $0.65977 \pm 0.00125$ | $0.66203 \pm 0.04319$ | $0.72264 \pm 0.00213$ | $0.69762 \pm 0.00654$ |
| 20 | $0.72997 \pm 0.00066$ | $0.72232 \pm 0.00103$ | $0.72624 \pm 0.00330$ | $0.72085 \pm 0.00106$ | $0.70252 \pm 0.01985$ | $0.72668 \pm 0.00307$ | $0.71395 \pm 0.00467$ |
| 25 | $0.73281 \pm 0.00030$ | $0.72339 \pm 0.00042$ | $0.73072 \pm 0.00193$ | $0.73253 \pm 0.00091$ | $0.71764 \pm 0.00987$ | $0.73084 \pm 0.00070$ | $0.72057 \pm 0.00444$ |
| 30 | $0.73420 \pm 0.00046$ | $0.72622 \pm 0.00049$ | $0.73425 \pm 0.00081$ | $0.73390 \pm 0.00026$ | $0.72267 \pm 0.00663$ | $0.72988 \pm 0.00434$ | $0.72487 \pm 0.00223$ |
| 35 | $0.73495 \pm 0.00040$ | $0.73225 \pm 0.00024$ | $0.73058 \pm 0.00350$ | $0.73512 \pm 0.00058$ | $0.73029 \pm 0.00509$ | $0.73361 \pm 0.00037$ | $0.73078 \pm 0.00102$ |

Table 8: Log-loss of Criteo experiments. SA is Sequential Attention, GL is generalized LASSO, and SL is Sequential LASSO. The values in the header for the LASSO methods are the $\ell_1$ regularization strengths used for each method.

| $k$ | SA | CMIM | GL ($\lambda = 10^{-1}$) | GL ($\lambda = 10^{-4}$) | SL ($\lambda = 10^{-1}$) | SL ($\lambda = 10^{-4}$) | Liao et al. (2021) |
|---|---|---|---|---|---|---|---|
| 5 | $0.14123 \pm 0.00005$ | $0.14323 \pm 0.00010$ | $0.14036 \pm 0.00046$ | $0.14519 \pm 0.00000$ | $0.14375 \pm 0.00163$ | $0.14073 \pm 0.00061$ | $0.14519 \pm 0.00146$ |
| 10 | $0.13883 \pm 0.00009$ | $0.13965 \pm 0.00008$ | $0.13747 \pm 0.00015$ | $0.14339 \pm 0.00019$ | $0.14263 \pm 0.00304$ | $0.13826 \pm 0.00032$ | $0.14082 \pm 0.00011$ |
| 15 | $0.13671 \pm 0.00007$ | $0.13745 \pm 0.00008$ | $0.13693 \pm 0.00005$ | $0.14227 \pm 0.00021$ | $0.14166 \pm 0.00322$ | $0.13713 \pm 0.00021$ | $0.13947 \pm 0.00050$ |
| 20 | $0.13633 \pm 0.00008$ | $0.13726 \pm 0.00010$ | $0.13693 \pm 0.00057$ | $0.13718 \pm 0.00004$ | $0.13891 \pm 0.00187$ | $0.13672 \pm 0.00035$ | $0.13806 \pm 0.00048$ |
| 25 | $0.13613 \pm 0.00013$ | $0.13718 \pm 0.00009$ | $0.13648 \pm 0.00051$ | $0.13604 \pm 0.00004$ | $0.13760 \pm 0.00099$ | $0.13628 \pm 0.00010$ | $0.13756 \pm 0.00043$ |
| 30 | $0.13596 \pm 0.00001$ | $0.13685 \pm 0.00004$ | $0.13593 \pm 0.00015$ | $0.13594 \pm 0.00005$ | $0.13751 \pm 0.00095$ | $0.13670 \pm 0.00080$ | $0.13697 \pm 0.00015$ |
| 35 | $0.13585 \pm 0.00002$ | $0.13617 \pm 0.00006$ | $0.13666 \pm 0.00073$ | $0.13580 \pm 0.00012$ | $0.13661 \pm 0.00096$ | $0.13603 \pm 0.00010$ | $0.13635 \pm 0.00005$ |

### B.4 The role of adaptivity

We show in this section the effect of varying adaptivity on the quality of selected features in Sequential Attention. In the following experiments, we select 64 features on six datasets by selecting $2^i$ features at a time over a fixed number of epochs of training, for $i \in \{0, 1, 2, 3, 4, 5, 6\}$. That is, we investigate the following question: for a fixed budget of training epochs, what is the best way to allocate the training epochs over the rounds of the feature selection process? For most datasets, we find that feature selection quality decreases as we select more features at once. An exception is the mice protein dataset, which exhibits the opposite trend, perhaps indicating that the features in the mice protein dataset are less redundant than in other datasets. Our results are summarized in Table 8 and Table 9. We also illustrate the effect of adaptivity for Sequential Attention on MNIST in Figure 9. One observes that the selected pixels "clump together" as $i$ increases, indicating a greater degree of redundancy.

Our empirical results in this section suggest that adaptivity greatly enhances the quality of features selected by Sequential Attention, and in feature selection algorithms more broadly.

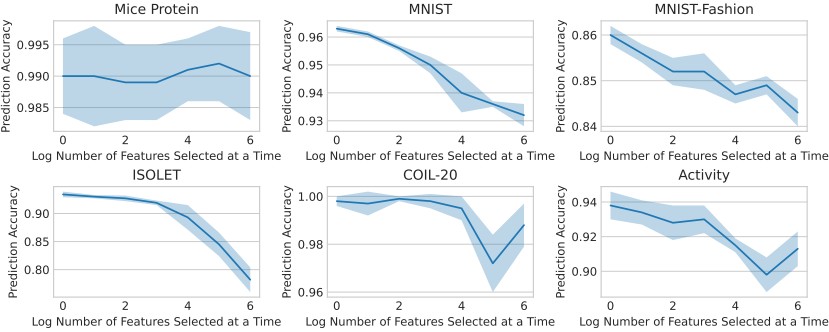

Figure 8: Sequential Attention with varying levels of adaptivity. We select 64 features for each model, and take $2^i$ features in each round for increasing values of $i$. We plot accuracy as a function of $i$.

Table 9: Sequential Attention with varying levels of adaptivity. We select 64 features for each model, and take $2^i$ features in each round for increasing values of $i$. We give the accuracy as a function of $i$.

| Dataset | $i = 0$ | $i = 1$ | $i = 2$ | $i = 3$ | $i = 4$ | $i = 5$ | $i = 6$ |
|---|---|---|---|---|---|---|---|
| Mice Protein | $0.990 \pm 0.006$ | $0.990 \pm 0.008$ | $0.989 \pm 0.006$ | $0.989 \pm 0.006$ | $0.991 \pm 0.005$ | $0.992 \pm 0.006$ | $0.990 \pm 0.007$ |
| MNIST | $0.963 \pm 0.001$ | $0.961 \pm 0.001$ | $0.956 \pm 0.001$ | $0.950 \pm 0.003$ | $0.940 \pm 0.007$ | $0.936 \pm 0.001$ | $0.932 \pm 0.004$ |
| MNIST-Fashion | $0.860 \pm 0.002$ | $0.856 \pm 0.002$ | $0.852 \pm 0.003$ | $0.852 \pm 0.004$ | $0.847 \pm 0.002$ | $0.849 \pm 0.002$ | $0.843 \pm 0.003$ |
| ISOLET | $0.934 \pm 0.005$ | $0.930 \pm 0.003$ | $0.927 \pm 0.005$ | $0.919 \pm 0.004$ | $0.893 \pm 0.022$ | $0.845 \pm 0.021$ | $0.782 \pm 0.022$ |
| COIL-20 | $0.998 \pm 0.002$ | $0.997 \pm 0.005$ | $0.999 \pm 0.001$ | $0.998 \pm 0.003$ | $0.995 \pm 0.005$ | $0.972 \pm 0.012$ | $0.988 \pm 0.009$ |
| Activity | $0.938 \pm 0.008$ | $0.934 \pm 0.007$ | $0.928 \pm 0.010$ | $0.930 \pm 0.008$ | $0.915 \pm 0.004$ | $0.898 \pm 0.010$ | $0.913 \pm 0.010$ |

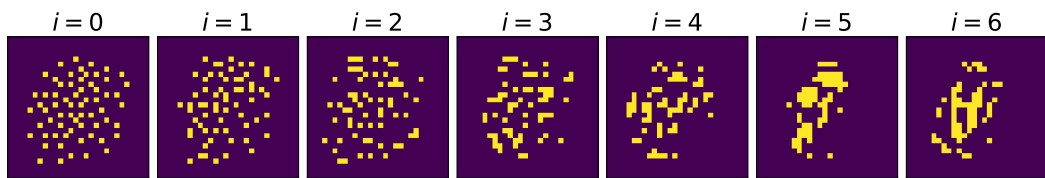

Figure 9: Sequential Attention with varying levels of adaptivity on the MNIST dataset. We select 64 features for each model, and select $2^i$ features in each round for increasing values of $i$.

### B.5 VARIATIONS ON HADAMARD PRODUCT PARAMETERIZATION

We provide evaluations for different variations of the Hadamard product parameterization pattern as described in Section 4.2. In Table 10, we provide the numerical values of the accuracies achieved.

Table 10: Accuracies of Sequential Attention for different Hadamard product parameterizations.

| Dataset | Softmax | $\ell_1$ | $\ell_2$ | $\ell_1$ normalized | $\ell_2$ normalized |
|---|---|---|---|---|---|
| Mice Protein | $0.990 \pm 0.006$ | $0.993 \pm 0.010$ | $0.993 \pm 0.010$ | $0.994 \pm 0.006$ | $0.988 \pm 0.008$ |
| MNIST | $0.958 \pm 0.002$ | $0.957 \pm 0.001$ | $0.958 \pm 0.002$ | $0.958 \pm 0.001$ | $0.957 \pm 0.001$ |
| MNIST-Fashion | $0.850 \pm 0.002$ | $0.843 \pm 0.004$ | $0.850 \pm 0.003$ | $0.853 \pm 0.001$ | $0.852 \pm 0.002$ |
| ISOLET | $0.920 \pm 0.003$ | $0.894 \pm 0.014$ | $0.908 \pm 0.009$ | $0.921 \pm 0.003$ | $0.921 \pm 0.003$ |
| COIL-20 | $0.997 \pm 0.004$ | $0.997 \pm 0.004$ | $0.995 \pm 0.006$ | $0.996 \pm 0.005$ | $0.996 \pm 0.004$ |
| Activity | $0.922 \pm 0.005$ | $0.906 \pm 0.015$ | $0.908 \pm 0.012$ | $0.933 \pm 0.010$ | $0.935 \pm 0.007$ |

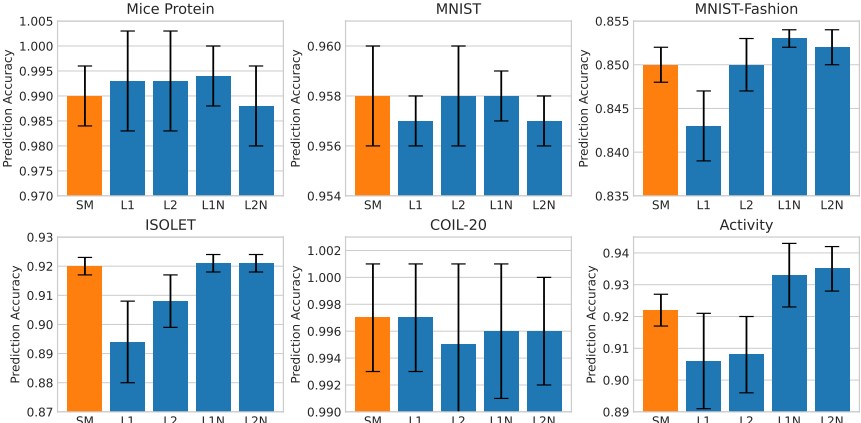

Figure 10: Accuracies of Sequential Attention for different Hadamard product parameterization patterns. Here, SM = softmax, L1 = $\ell_1$, L2 = $\ell_2$, L1N = $\ell_1$ normalized, and L2N = $\ell_2$ normalized.

## B.6 APPROXIMATION OF MARGINAL GAINS

Finally, we present our experimental results that show the correlations between weights computed by Sequential Attention and traditional feature selection *marginal gains*.

**Definition B.1** (Marginal gains). Let $\ell : 2^{[d]} \to \mathbb{R}$ be a loss function defined on the ground set $[d]$. Then, for a set $S \subseteq [n]$ and $i \notin S$, the *marginal gain of $i$ with respect to $S$ is $\ell(S) - \ell(S \cup \{i\})$*.

In the setting of feature selection, marginal gains are often considered for measuring the importance of candidate features $i$ given a set $S$ of features that have already be selected by using the set function $\ell$, which corresponds to the model loss when trained on a subset of features. It is known that greedily selecting features based on their marginal gains performs well in both theory (Das & Kempe, 2011; Elenberg et al., 2018) and practice (Das et al., 2022). These scores, however, can be extremely expensive to compute since they require training a model for every feature considered.

In this experiment, we first compute the top $k$ features selected by Sequential Attention for $k \in \{0, 9, 49\}$ on the MNIST dataset. Then we compute (1) the true marginal gains and (2) the attention weights according to Sequential Attention, conditioned on these features being in the model. The Sequential Attention weights are computed by only applying the attention softmax mask over the $d - k$ unselected features, while the marginal gains are computed by explicitly training a model for each candidate feature to be added to the preselected $k$ features. Because our Sequential Attention algorithm is motivated by an efficient implementation of the greedy selection algorithm that uses marginal gains (see Section 1), one might expect that these two sets of scores are correlated in some sense. We show this by plotting the top scores according to the two sets of scores and by computing the Spearman correlation coefficient between the marginal gains and attention logits.

In the first and second rows of Figure 11, we see that the top 50 pixels according to the marginal gains and attention weights are visually similar, avoiding previously selected regions and finding new areas which are now important. In the third row, we quantify their similarity via the Spearman correlation between these feature rankings. While the correlations degrade as we select more features (which is to be expected), the marginal gains become similar among the remaining features after removing the most important features.

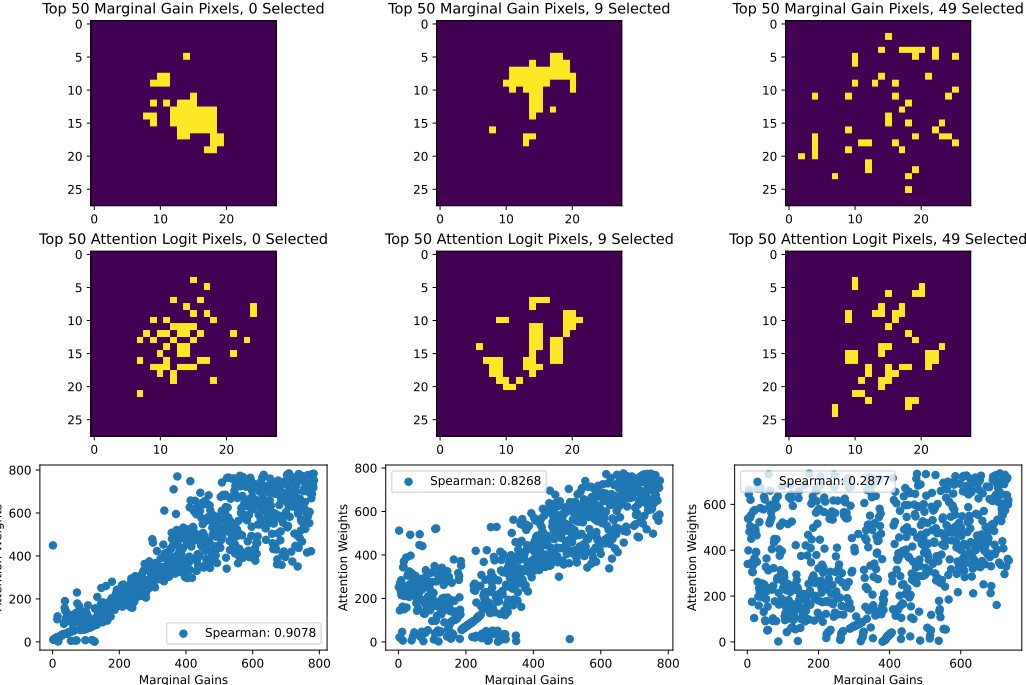

Figure 11: Marginal gain experiments. The first and second rows show that top 50 features chosen using the true marginal gains (top) and Sequential Attention (middle). The bottom row shows the Spearman correlation between these two computed sets of scores.

