# OpenReview forum: "Sequential Attention for Feature Selection"
_ICLR.cc/2023/Conference — ICLR 2023 poster_

### Official Review · Reviewer_vzVg · 2022-10-18

**Confidence:** 4
**Correctness:** 3
**Technical Novelty And Significance:** 2
**Empirical Novelty And Significance:** 3
**Recommendation:** 5

**Clarity, Quality, Novelty And Reproducibility:**

Reproducibility:
* The algorithm is easy to reproduce. However, some hyper-parameters appears to be missing. The value for $\lambda$ in the sequential LASSO is not explained. Some information regarding the training procedure (number of epochs, optimizer, ...) is also missing.

Novelty:
* The use of a softmax mask attached to the input was previously discussed in [1]. It is worth mentioning the authors reduce the matrix mask presented in [1] to a binary vector, but it increases its computational cost. Other approaches like DFS [2] or SFS [3] provide a similar structure, using only one extra vector that multiplies the input data.

Clarity:
* The algorithm is easy to understand and implement. The theoretical explanation is also interesting, adding extra value to the contribution.

Quality:
* The experimental section is poor. The authors only compared their algorithm against other incremental FS algorithms. Although it is a good sign to establish similar or better results in this specific approach, there exists multiple FS algorithms with better results than the ones provided in the paper. There is a lack of comparison against similar FS architectures like CAE [1], DFS[2] or SFS[3].

* The use of the softmax function over the mask can be difficult when the number of features is very high. It could case the mask initialization to have values extremely close to zero. This could led to a difficult mask training, as the gradient over those vector has also close to zero values. I suggest the authors to maybe include a temperature parameter to their softmax function.



[1] Balın, M. F., Abid, A., & Zou, J. (2019, May). Concrete autoencoders: Differentiable feature selection and reconstruction. In International conference on machine learning (pp. 444-453). PMLR.

[2] Zou, Q., Ni, L., Zhang, T., & Wang, Q. (2015). Deep learning based feature selection for remote sensing scene classification. IEEE Geoscience and Remote Sensing Letters, 12(11), 2321-2325.

[3] Cancela, B., Bolón-Canedo, V., Alonso-Betanzos, A., & Gama, J. (2020). A scalable saliency-based feature selection method with instance-level information. Knowledge-Based Systems, 192, 105326.

**Strength And Weaknesses:**

Strengths:
* The idea is easy to implement
* The theoretical explanation is interesting
* The spatial complexity of the algorithm is low

Weaknesses:
* The idea is somehow trivial
* The algorithm time complexity is extremely high
* The experimental results only cover other incremental Feature Selection methods.


**Summary Of The Paper:**

The authors propose a sequential algorithm for feature selection, where a feature is selected in each step. The intuition behind the idea is to minimize the selection of redundant features. A theoretical study is also presented, matching the proposed algorithm with other state-of-the-art methods like OMP or Sequential LASSO. The experimental results show promising accuracy scores when compared against other incremental Feature Selection methods.

**Summary Of The Review:**

Overall, I think it is an interesting solution, but more work has to be done before publishing it. More comparisons against other FS algorithms have to be performed. Besides that, I think the novel contribution of this algorithm is questionable, as the solution is extremely similar to the mentioned OMP and sequential LASSO.

---

> ### Author Response · Authors · 2022-11-16
> **Thank you for the review (1/2)**
>
> We thank Reviewer vzVg for recognizing the simplicity of our algorithm, our theoretical contributions, and the space efficiency of our method. We address some of the comments below:
>
> > The idea is somehow trivial
>
> We view the simplicity of our algorithm as a major strength of our algorithm, rather than a weakness. While past literature considered complicated algorithms that use techniques such as autoencoders (DFS, CAE) and aggregated instance-wise saliency scores (SFS, Liao-Latty-Yang), we empirically outperform these methods with a simple algorithm and much lower overhead by just introducing just $d$ new variables for the global softmax mask together with forward selection.
>
> > The algorithm time complexity is extremely high
>
> We agree that, as formulated in Algorithm 1, the running time of the algorithm would be high due training $k$ models. However, the fact that the algorithm trains $k$ models is only to facilitate the presentation of our algorithm, and in our practical implementations, we in fact optimize our algorithm to run during *one* model training. We do so by partitioning the total training epochs into $k$ parts, and selecting one feature in each part. We note that the decreased epochs for each model training does not significantly affect the quality of selected features, as much fewer epochs are required if one only wishes to identify important features, rather than fully training a predictive model. This optimization is discussed in Appendix B.6 of our original draft and in Section 4 of our revised draft. Due to this optimization and the lower space complexity of the new trainable variables, we believe that our algorithm is far more efficient in running time than previously proposed algorithms. Based on this review, we added a comprehensive discussion and analysis of the resource efficiencies of the different methods in Appendix B.2.3 of our revised draft by evaluating the running time and prediction accuracy of Sequential Attention compared to all other baseline algorithms while using a fixed (total) number of epochs and batch size. Our results show that Sequential Attention offers the highest prediction accuracy while using the least amount of training (wall) time. For instance, on MNIST, Sequential Attention finds 50 features achieving 0.958 accuracy on the test set in under a minute.
>
> > The experimental results only cover other incremental Feature Selection methods.
>
> In addition to incremental feature selection methods such as Sequential LASSO and OMP, we also considered one-round ranking-based methods such as Group LASSO and the attention-based method of Liao-Latty-Yang. In the revision, we also added a comparison to the Concrete Autoencoder (CAE), as suggested by the reviewer. We note that our choice of dataset and neural network architecture also allow for a comparison to the experiments considered in [LRT2021], which includes a comparison to several other feature selection methods, as detailed in Appendix B.2.1 of our original draft.
>
> [LRT2021] Ismael Lemhadri, Feng Ruan, and Rob Tibshirani. Lassonet: Neural networks with feature sparsity. In International Conference on Artificial Intelligence and Statistics, pp. 10–18. PMLR, 2021.
>
> > However, some hyper-parameters appears to be missing. The value for λ in the sequential LASSO is not explained. Some information regarding the training procedure (number of epochs, optimizer, ...) is also missing.
>
> We commented on this in our initial response to all authors, but we are also repeating it here to keep things organized:
>
> We provide the full details of our feature selection training procedure and hyperparameters in our colab notebook demo. Regarding the choice of λ in Sequential LASSO and Group LASSO, we use one value of λ (`group_lasso_scale` in the colab notebook) as the regularization strength for all $k$ steps of the sequential feature selection process. This λ parameter is then tuned to give the best overall feature selection results.

---

> ### Author Response · Authors · 2022-11-16
> **Thank you for the review (2/2)**
>
> > The use of a softmax mask attached to the input was previously discussed in [1]. It is worth mentioning the authors reduce the matrix mask presented in [1] to a binary vector, but it increases its computational cost. Other approaches like DFS [2] or SFS [3] provide a similar structure, using only one extra vector that multiplies the input data.
>
> While we use a softmax mask in the Sequential Attention algorithm, this is not where our central piece of novelty lies. Indeed, we recognize and point out several works based on using softmax masks for feature selection, such as [LLY2021] which aggregates instance-wise scores to construct global feature importance scores just as SFS [3] does, amidst other related works. Rather, our main point of novelty is in (1) applying attention-based feature selection sequentially to capture the residual value of features given the currently selected set, and (2) directly learning importance scores for each feature instead of aggregating instance-wise features. This simplifies the algorithm and model architecture, and allows us to obtain provable guarantees. In particular, DFS [2] and SFS [3] both are based on aggregating instance-wise scores and do not make adaptive decisions to capture the residual effect of features.
>
> [LLY2021] Yiwen Liao, Raphael Latty, and Bin Yang. Feature selection using batch-wise attenuation and feature mask normalization. In 2021 International Joint Conference on Neural Networks (IJCNN), pp. 1–9. IEEE, 2021.
>
> > The experimental section is poor. The authors only compared their algorithm against other incremental FS algorithms. Although it is a good sign to establish similar or better results in this specific approach, there exists multiple FS algorithms with better results than the ones provided in the paper. There is a lack of comparison against similar FS architectures like CAE [1], DFS[2] or SFS[3].
>
> We thank the reviewer for these references. As previously discussed, our results are compared to a number of methodologies beyond just incremental feature selection algorithms.
>
> We reiterate our response from our initial response:
>
> We provide an explicit comparison to CAE in our colab notebook (using the CAE implementation provided by the authors), as well as in Section 4.1 of our revised draft. The results show that Sequential Attention is superior to CAE for all `num_selected_features` considered.  For instance, on MNIST, our experiments show that Sequential Attention finds a set of $k = 50$ features with a prediction accuracy that is 5% better than CAE, in half the selection time. We note the DFS algorithm is quite similar to CAE, so the performance of CAE is likely representative since both methods seek to select a small subset of features in an unsupervised manner by learning a sparse autoencoder that depends on only a few features. We will add citations/discussions for these works and the CAE evaluations to our experiment figures in Section 4.1.
>
> We believe our new comparison to the LLY algorithm is representative of the SFS algorithm. Both algorithms are based on the idea of aggregating instance-wise feature importance scores (saliency) to give a global feature selection algorithm. The methods for obtaining this saliency are also similar — SFS uses a softmax score and LLY uses a similar attention-based score. As stated in our paper, we believe our main improvements over works such as LLY and SFS are (1) a simpler training process that directly trains a global feature importance score vector via a softmax mask instead of aggregating instance-wise information, and (2) our use of adaptivity to identify a diverse set of high-signal features.
>
> Lastly, the Group LASSO results in the initial draft are a one-shot algorithm (i.e., no adaptivity), and has noticeably worse performance than Sequential LASSO.
>
> > The use of the softmax function over the mask can be difficult when the number of features is very high. It could case the mask initialization to have values extremely close to zero. This could led to a difficult mask training, as the gradient over those vector has also close to zero values. I suggest the authors to maybe include a temperature parameter to their softmax function.
>
> We agree with the reviewer that a temperature parameter is important in the training process. In fact, the temperature parameter is equivalent to the learning rate of the training of the softmax attention logits, and we have tuned their learning rate in our experiments. In the revision, we have emphasized that the temperature parameter is actually equivalent to the learning rate of the softmax attention logits.

---

### Official Review · Reviewer_FwQ2 · 2022-10-23

**Confidence:** 3
**Correctness:** 3
**Technical Novelty And Significance:** 2
**Empirical Novelty And Significance:** 2
**Recommendation:** 6

**Clarity, Quality, Novelty And Reproducibility:**

The paper is well organized and written. The novelty of the paper mainly lies in the theoretical analysis of the connection between the sequential attention and existing approaches. The algorithm is clearly described and should be reproducible.

**Strength And Weaknesses:**

Strength
- The paper is well written and organized.
- The paper proposes efficient sequential feature selection methods with attention, which reduces the complexity of greedy forward based feature selection.
- The authors provide provable guarantees for Sequential Attention for least squares linear regression by analyzing its variation called regularized linear Sequential Attention. They show the connection between Sequential LASSO and Orthogonal Matching Pursuit.

Weakness
- The authors claimed a lot of merits for the proposed algorithms, such as “this technique reduces the overhead of our algorithm, removes the need to tune unnecessary hyperparameters, works directly with any model architecture,…achieves state-of-the-art feature selection results for neural networks on standard benchmarks”. However, the experiments seem not quite complete to support the claim. For example, there is no experiment demonstrating the approach works on all model architectures.
- The advantages over existing methods are not clear or significant. The results in Figure 3 on DNNs seems to suggest there is limited advantage over SL, GL or OMP for these datasets.
- The scale of the experiment on DNN is quite small and limited. The experiments on neural networks are done with only a one-layer neural network with hidden width 67 and ReLU activation.


**Summary Of The Paper:**

This work introduces the Sequential Attention algorithm for supervised feature selection. This algorithm is based on an efficient implementation of greedy forward selection and uses attention weights at each step as a proxy for marginal feature importance.
The authors provide theoretical insights into the Sequential Attention algorithm for linear regression models by showing that its regularized linear Sequential Attention model is equivalent to sequential LASSO and Orthogonal Matching Pursuit, and thus inherits all of their provable guarantees. Their theoretical and empirical analyses provide new explanations towards the effectiveness of attention and its connections to overparameterization.

**Summary Of The Review:**

The novelty mainly lies in theoretical analysis. The theoretical and experiments both suggest the proposed approach is similar to the existing approach and the benefits are not clear.

---

> ### Author Response · Authors · 2022-11-16
> **Thank you for the review (1/2)**
>
> We thank Reviewer FwQ2 for recognizing our theoretical analysis of Sequential Attention and its    implications on the effectiveness of using the attention mechanism for feature selection. We address some of the comments below:
>
> > The authors claimed a lot of merits for the proposed algorithms, such as “this technique reduces the overhead of our algorithm, removes the need to tune unnecessary hyperparameters, works directly with any model architecture,…achieves state-of-the-art feature selection results for neural networks on standard benchmarks”. However, the experiments seem not quite complete to support the claim. For example, there is no experiment demonstrating the approach works on all model architectures.
>
> Thank you for pointing this out to us. We would like to take this opportunity to highlight our discussions on the various advantages of our algorithm throughout our paper. We summarize the justification for our claims as follows. The reduction of overhead refers to our simplification of prior attention-based feature selection algorithms by directly training a softmax mask instead of aggregating instance-wise feature importances. In particular, our approach only introduces a single additional trainable variable for each feature, whereas prior approaches add a whole subnetwork to compute instance-wise feature importances. This also leads to removal of unnecessary hyperparameters — prior approaches introduce many additional hyperparameters for specifying the model architecture that learns instance-wise feature importances. In contrast, our algorithm has no additional hyperparameters. We agree with the reviewer that we have not provided extensive experimental evidence that Sequential Attention performs well with any model architecture. However, our theoretical analysis applies to the linear regression model, and our experimental results include two different scales of neural networks, including one small-scale network with one hidden layer with 67 neurons, and one large-scale network with three hidden layers with 768, 256, and 128 neurons. Furthermore, our algorithm makes sense for any differentiable model, as formalized in Algorithm 1. Our one-pass implementation of Sequential Attention is described in Appendix B.3 of the original draft, and is necessary for handling the Criteo experiment due to its size. Our investigation of the connection between Sequential Attention and marginal gains is discussed in more detail in Appendix B.6 of the original draft.
>
> > The advantages over existing methods are not clear or significant. The results in Figure 3 on DNNs seems to suggest there is limited advantage over SL, GL or OMP for these datasets.
>
> We believe Figure 3 strongly suggests the superiority of Sequential Attention compared to other methods. While SL, GL, and OMP can be competitive with Sequential Attention on certain datasets, Sequential Attention is a *single* algorithm that is *uniformly* competitive with or better than these algorithms, and is thus more reliable from a practitioner’s perspective. Concretely, Sequential Attention can serve as a single go-to algorithm for feature selection.

---

> ### Author Response · Authors · 2022-11-16
> **Thank you for the review (2/2)**
>
> > The scale of the experiment on DNN is quite small and limited. The experiments on neural networks are done with only a one-layer neural network with hidden width 67 and ReLU activation.
>
> We thank the reviewer for encouraging us to further discuss our results on large scale datasets. To showcase the scalability of our algorithm, we perform large-scale feature selection experiments on the Criteo dataset in the section titled Large-scale Experiments, which contains over *3 billion training examples*. In our revision, we have clarified the size of the neural network used for this pCTR experiment, which has three hidden layers with 768, 256, and 128 neurons.
>
> We also note that our small-scale experiments for feature selection are well-motivated by previous work, as they follow the experiment setup in previous works both in the choice of dataset and the the neural network architecture [ABZ2019, GGH2019, LLY2021, LRT2021]. We hope that by providing both (1) smaller scale experiments that are directly comparable to prior work, and (2) large scale experiments on the order of 3 billion training examples and hundreds of thousands of weights, we have investigated the capabilities of Sequential Attention for a wide range of applications.
>
> [ABZ2019] Muhammed Fatih Balın, Abubakar Abid, and James Zou. Concrete autoencoders: Differentiable feature selection and reconstruction. In International Conference on Machine Learning (ICML), pp. 444–453. PMLR, 2019.
>
> [GGH2019] Ning Gui, Danni Ge, and Ziyin Hu. AFS: An attention-based mechanism for supervised feature selection. In Proceedings of the AAAI conference on Artificial Intelligence (AAAI), volume 33, pp. 3705–3713, 2019.
>
> [LLY2021] Yiwen Liao, Raphael Latty, and Bin Yang. Feature selection using batch-wise attenuation and feature mask normalization. In 2021 International Joint Conference on Neural Networks (IJCNN), pp. 1–9. IEEE, 2021.
>
> [LRT2021] Ismael Lemhadri, Feng Ruan, and Rob Tibshirani. Lassonet: Neural networks with feature sparsity. In International Conference on Artificial Intelligence and Statistics, pp. 10–18. PMLR, 2021.

---

### Official Review · Reviewer_tvep · 2022-10-25

**Confidence:** 3
**Correctness:** 3
**Technical Novelty And Significance:** 3
**Empirical Novelty And Significance:** 3
**Recommendation:** 6

**Clarity, Quality, Novelty And Reproducibility:**

My biggest concern in this paper is regarding clarity. The experimental details are often unspecified or hard to understand. Here are some examples:

- $\bar{S}$ is not defined anywhere (even though it becomes evident from the paper what it means)
- Some tables are referenced as figures (e.g., figure 7)
- What do the shadowed regions represent in Figure 4?
- It is hard to understand the correlation plots in Figure 14. What was the dataset? Was the model trained until the end? Can you provide more details about this?
- How was the method implemented efficiently (as stated in the abstract)? What are the running speed and memory consumption? How do they compare to related approaches?
- It is unclear how the "empirical success of attention-based feature selection is primarily due to the explicit overparameterization". The additional parameters are only $d$, a tiny number, even for the experimented datasets.
- The examples of selected features are very uninformative (probably because of the dataset). Displaying independent pixels is hard to interpret, so it is unclear if the subset of selected features is meaningful.

**Strength And Weaknesses:**

Things that I liked in this paper:
- The paper has a strong math foundation and excels at theoretical contributions.
- The proposed method outperforms related approaches despite being very simple.
- On top of this, this paper shows empirically and theoretically that "attention" is, in fact, a suitable approach to selecting a good subset of features.

However, I think the paper can be improved, especially regarding clarity. The paper focuses significantly on the theoretical aspect, while the experimental section is very short, with several important experiments left in the appendix. For example, it was not until section 4.2 that I realized that the previous results were achieved using the standard Sequential Attention shown in Algorithm 1. Finally, I believe that Algorithm 1, in its current form, is very inefficient since it requires training at least $k$ models.


**Summary Of The Paper:**

This paper proposes a method to perform feature selection by learning to weigh the input features with probabilities given by a softmax transformation. Given the similarity to attention methods, the method is called Sequential Attention. It consists of running the model with the reminiscent non-selected features and then greedly selecting the features associated with the highest softmax weight. The authors provide a large body of theoretical guarantees, linking an adapted (regularized) version of Sequential Attention to the classical Sequential LASSO algorithm. Results on 7 datasets showcase the predictive performance of the proposed approach.

**Summary Of The Review:**

Overall this paper provides a great theoretical contribution to attention-based feature selection. It also presents solid empirical results on several datasets, outperforming related approaches. However, I believe it has room for improvement, especially regarding the clarity of the experimental setup.

---

> ### Author Response · Authors · 2022-11-16
> **Thank you for the review (1/3)**
>
> We thank Reviewer tvep for supporting our theoretical contributions and acknowledging the simplicity and strong practical performance of our algorithm. We address some of the comments below:
>
> > The paper focuses significantly on the theoretical aspect, while the experimental section is very short, with several important experiments left in the appendix.
>
> As noted by the current reviewer and other reviewers, our novel analysis of Sequential Attention and Sequential LASSO, as well as the various conceptual implications of our proofs, are one of our strongest contributions, so we devote a large portion of our work towards this discussion. While we demonstrate empirical improvements over previous works in our Experiments section, strong experiments are just one of the contributions we make and not our only focus.
>
> In particular, some other conceptual takeaways beyond strong empirical performance that we highlight are (1) the value of sequential algorithms for feature selection for neural networks (2) the role of attention weights as a regularizer of model weights, and (3) the equivalence between Sequential LASSO and OMP, which connects sequential regularization with marginal gains. We are happy to take suggestions on which experiments are best-suited to be in the main body of the paper.
>
> > For example, it was not until section 4.2 that I realized that the previous results were achieved using the standard Sequential Attention shown in Algorithm 1.
>
> We made this clear in the revised version by explicitly stating at the beginning of Section 4 that our experimental results use the Sequential Attention version as presented in Algorithm 1.
>
> > Finally, I believe that Algorithm 1, in its current form, is very inefficient since it requires training at least k models.
>
> We agree with the reviewer that training $k$ models would be very inefficient. However, in our practical implementation of Sequential Attention, we circumvent this problem and efficiently implement the algorithm so that it selects features over only *one model training*. As noted in Appendix B.3 of our paper (before revision), we implement our algorithm to run during one model training by partitioning the training epochs into $k$ parts and selecting a feature in each of these $k$ parts. We provide the implementation of this “one-pass” version in the colab demo. We note that the decreased epochs for each model training does not significantly affect the quality of selected features, as much fewer epochs are required if one only wishes to identify important features, rather than fully training a predictive model.
>
> In the revised version, we moved the discussion of this optimization to the main body of the paper in Section 4. We also added a section in the appendix (Appendix B.2.3) detailing these notes on efficiency, together with measurements of the running time for Sequential Attention and comparison to other baseline methods. Our empirical results indicate that for a fixed budget of training epochs, Sequential Attention obtains the highest accuracy results and lowest wall clock times. For instance, on MNIST, Sequential Attention finds 50 features achieving 0.958 accuracy on the test set in under a minute.
>
> > My biggest concern in this paper is regarding clarity. The experimental details are often unspecified or hard to understand.
>
> We provided a full implementation of our experiments for MNIST in our colab notebook, which can be found in our response to all reviewers. We will make the experimental setup clearer (e.g., model architectures, choice of hyperparameters) in the revised version. We hope this clarifies any details that the reviewer has found underspecified.
>
> > S¯ is not defined anywhere (even though it becomes evident from the paper what it means)
>
> We thank the reviewer for pointing this out. We updated Algorithm 1 of the revised draft to include this definition.
>
> > Some tables are referenced as figures (e.g., figure 7)
>
> Thank you for catching this — we fixed every instance of this problem in the revised draft.
>
> > What do the shadowed regions represent in Figure 4?
>
> The shadowed regions linearly interpolate the standard deviation error bounds, and was explained in Appendix B.3 of our original draft. We updated Appendix B.3 to make this more clear.

---

> > ### Comment · Reviewer_tvep · 2022-11-21
> > **Response**
> >
> > I want to thank the authors for the clarifications and the detailed and thorough responses.
> >
> >
> > _On the meaningfulness of the selected features:_ Thanks for the clarifications. I believe that these clarifications, even if informal, can potentially help the interested readers.
> >
> >
> > > we implement our algorithm to run during one model training by partitioning the training epochs into $k$ parts and selecting a feature in each of these $k$ parts.
> >
> > This strategy definitely solves the issue. However, it is a greedy approximation: the selection on timestep $t+1$ will be influenced by the features selected on timestep $t$. So, what is the gap to actually training $k$ models?
> >
> >
> > Given that most of my concerns were properly addressed and the authors considered the reviewers' suggestions, I am happy to increase my score.

---

> > > ### Author Response · Authors · 2022-11-28
> > > **Thank you for your response**
> > >
> > > We thank the reviewer for their response and increasing their score. We will include additional discussion about the interpretation of the visualized pixels on MNIST in our next revision. Concerning the “one model training” implementation of our algorithm, the gap between our efficient implementation and fully training $k$ models is just a trade-off between the total number of epochs dedicated to feature selection and the resulting quality of the selected features. Indeed, if the total number of epochs is set to be $k$ times the number of epochs for (the original) one model training, this recovers the simple algorithm of fully training $k$ models. When using a smaller number of total epochs, we shuffle our dataset to approximate the “$k$ model training” version of our algorithm. We empirically observe that beyond a small number of epochs, the quality of selected features does not differ significantly, and that these small numbers of epochs already deliver state-of-the-art quality of feature selection. We empirically demonstrate trade-offs between the number of epochs used to select features and resulting prediction accuracy in a new colab notebook hosted here: https://colab.research.google.com/drive/12flu35EsB_mEXkJ1epbXlsGVKcyC1_-R.

---

> > > > ### Comment · Reviewer_tvep · 2022-12-02
> > > > **Comment**
> > > >
> > > > > if the total number of epochs is set to be $k$ times the number of epochs for (the original) one model training, this recovers the simple algorithm of fully training $k$ models.
> > > >
> > > > This is only true if the $(k+1)$th model is initialized with the weights learned by the $k$th model. Therefore, there is a gap between the theoretical algorithm and its practical implementations. My question is how large is this gap in terms of feature quality?
> > > >
> > > > > When using a smaller number of total epochs, we shuffle our dataset to approximate the “ model training” version of our algorithm
> > > >
> > > > I did not understand how this could solve the above issue. Can you clarify?

---

> > > > > ### Author Response · Authors · 2022-12-06
> > > > > **Thank you for your comment**
> > > > >
> > > > > > This is only true if the $(k+1)$th model is initialized with the weights learned by the $k$th model. Therefore, there is a gap between the theoretical algorithm and its practical implementations. My question is how large is this gap in terms of feature quality?
> > > > >
> > > > > Thank you for the question. As the reviewer pointed out, there is a difference in whether or not the weights are reset when comparing the “fully-retrained version” vs the “one-pass implementation.” In the experiments presented in the paper, we *did* reset the attention weights, but *did not* reset the neural network weights. We provide a new notebook (https://colab.sandbox.google.com/drive/1R_31YKhfVb1huvzn1pRgBGo05vDVUI7e) with an additional experiment showing that resetting the neural network weights does not make a significant difference to the performance of Sequential Attention, except for the MNIST-Fashion dataset, for which resetting all neural network weights in each round resulted in a slight degradation in feature selection quality.
> > > > >
> > > > > Our implementation of Sequential Attention also included a “warm start” phase in each selection step, where the neural network weights were pretrained with a small amount of data (represented by the configurable parameter `start_percentage = 0.05` in the first colab demo). We also investigated the effects of this warm starting in the follow-up experiment, and found that it can slightly help the prediction accuracy at times and hurt it at others. Despite these modifications to Algorithm 1 in our paper, we believe our theoretical analysis captures a large fraction of the explanation behind the success of Sequential Attention. These small changes (knobs) are just practical generalizations that may or may not help.
> > > > >
> > > > > > I did not understand how this could solve the above issue. Can you clarify?
> > > > >
> > > > > We agree that shuffling does not address whether or not the model weights are reset. The shuffling just ensures that the $k$ different phases of the one pass training have a roughly similar distribution of training examples since feature selection spans multiple epochs.

---

> ### Author Response · Authors · 2022-11-16
> **Thank you for the review (2/3)**
>
> > It is hard to understand the correlation plots in Figure 14. What was the dataset? Was the model trained until the end? Can you provide more details about this?
>
> We thank the reviewer for pointing out the lack of clarity in Appendix B.6 and Figure 14 in the original draft. The goal of this section is to explore the correlation between attention weights computed in our Sequential Attention algorithm and marginal gains, which is defined for a candidate feature $i$ and currently selected features $S$ as the difference in the model loss when trained on features $S$ versus the features $S\cup\{i\}$. We conduct our experiments on the MNIST dataset. Here, we first compute a set $S_k$ of $k$ features using Sequential Attention, for $k\in\{0,9,49\}$. Then, for the remaining features $i\in[d]\setminus S_k = \overline{S}_k$, we compute Sequential Attention weights and marginal gates conditioned on the selected features $S_k$. For Sequential Attention, this means that the softmax mask is only applied outside of the set $S_k$ of preselected features. The first two rows of Figure 14 (in the original draft) are visualizations of the top 50 pixels with the highest Sequential Attention weight and marginal gains, respectively, while the third row displays the Spearman correlation between these two scores. The number of epochs used to train the Sequential Attention weights are the same as what we use in our end-to-end feature selection results in the Experiments section, and the number of epochs used to train the model to compute marginal gains is the same as what we use to retrain the model on a subset of weights in the Experiments section. Our new revised draft includes more details about this experiment.
>
> > How was the method implemented efficiently (as stated in the abstract)? What are the running speed and memory consumption? How do they compare to related approaches?
>
> We discuss how our Sequential Attention algorithm can be viewed as an efficient implementation of the greedy algorithm in the paragraph titled “Sequential Attention” on page 2 of our original draft. The essence of the idea is that in the greedy algorithm, one trains $d$ models for every feature that is selected, where $d$ is the total number of feature candidates. Our Sequential Attention implements this idea more efficiently by using the attention mechanism to evaluate $d$ feature candidates using only one model training.
>
> As addressed earlier, we have added running time evaluations in Appendix B.2.3 of our revised draft. Concerning the memory overhead, our algorithm introduces just $d$ additional trainable variables to the model architecture, where $d$ is the number of inputs to the original model. This is far more efficient than other proposed methods, which often consider adding a whole separate neural network to the model architecture, whose first layer would already have a higher memory and training overhead than our proposed algorithm. As just one example, the Liao-Latty-Yang attention algorithm adds a neural network which outputs a softmax mask which yields instance-wise saliency scores which are aggregated to form a global feature ranking score, whereas we add just $d$ trainable weights.

---

> ### Author Response · Authors · 2022-11-16
> **Thank you for the review (3/3)**
>
> > It is unclear how the "empirical success of attention-based feature selection is primarily due to the explicit overparameterization". The additional parameters are only d, a tiny number, even for the experimented datasets.
>
> We thank the reviewer for pointing out this confusion. Our use of the term “overparameterization” refers to the addition of trainable variables which serve as attention weights, which is just $d$ additional variables for a model with $d$ inputs. We use this terminology to conform to the notion of “Hadamard product overparameterization” [VKR2019], which refers to this type of addition of variables. This differs from another line of work on overparameterization for neural networks, which refers to the addition of a massive number of variables which far exceeds even the number of training examples. We apologize for the confusion, and we have clarified this in our revised draft.
>
> In our theoretical analysis of regularized linear Sequential Attention, the only significant modification we make to standard linear regression is the addition of a Hadamard product with an additional set of variables, which correspond to attention weights (which we refer to as “Hadamard product overparameterization”). As we prove, this is already sufficient to provably explain how an informative set of features can be identified. We believe that this theoretical analysis captures the heart of how attention-based methods succeed in identifying informative features, and thus we attribute the success of attention-based feature selection methods to this addition of the Hadamard product overparameterization.
>
> [VKR2019] Tomas Vaskevicius, Varun Kanade, and Patrick Rebeschini. Implicit regularization for optimal sparse recovery. Advances in Neural Information Processing Systems, 32, 2019.
>
> > The examples of selected features are very uninformative (probably because of the dataset). Displaying independent pixels is hard to interpret, so it is unclear if the subset of selected features is meaningful.
>
> We agree that the interpretation of the “meaningfulness” of the selected features is subjective, and we make no formal claims concerning what they indicate or what information they capture. However, pixels selected on MNIST have been visualized in many prior works [ABZ2019, GGH2019, LLY2021, LRT2021] on feature selection, and serves as a basic sanity check of what kinds of pixels are being selected by feature selection algorithms. For instance, the degree to which the selected pixels “cluster” can be used to informally assess the redundancy of features selected on image datasets, since neighboring pixels tend to represent redundant information. It is also at times useful to assess which “regions” of the image are selected. For example, the central regions of images tend to be more informative than the edges of images.
>
> [ABZ2019] Muhammed Fatih Balın, Abubakar Abid, and James Zou. Concrete autoencoders: Differentiable feature selection and reconstruction. In International Conference on Machine Learning (ICML), pp. 444–453. PMLR, 2019.
>
> [GGH2019] Ning Gui, Danni Ge, and Ziyin Hu. AFS: An attention-based mechanism for supervised feature selection. In Proceedings of the AAAI conference on Artificial Intelligence (AAAI), volume 33, pp. 3705–3713, 2019.
>
> [LLY2021] Yiwen Liao, Raphael Latty, and Bin Yang. Feature selection using batch-wise attenuation and feature mask normalization. In 2021 International Joint Conference on Neural Networks (IJCNN), pp. 1–9. IEEE, 2021.
>
> [LRT2021] Ismael Lemhadri, Feng Ruan, and Rob Tibshirani. Lassonet: Neural networks with feature sparsity. In International Conference on Artificial Intelligence and Statistics, pp. 10–18. PMLR, 2021.

---

### Official Review · Reviewer_ABqs · 2022-10-25

**Confidence:** 3
**Correctness:** 4
**Technical Novelty And Significance:** 3
**Empirical Novelty And Significance:** 2
**Recommendation:** 8

**Clarity, Quality, Novelty And Reproducibility:**

Clearly written study, the methodology seems fairly novel.

Minor comments
Related work could have mentioned another relevant feature selection approach MRMR - Maximum Relevance Minimum Redundancy. This instance of greedy forward selection algorithm tends to select a subset of features having the most correlation with the target, but the least correlation wrt already selected features.
Section 4.2 there is redundant “Algorithm Algorithm 1”


**Strength And Weaknesses:**

Strengths
While equivalence of Sequential LASSO and OMP was shown under several assumptions before, this work shows the equivalence holds for more general settings.
Clever use of attention to achieve computational savings by evaluating unselected features all at once, at every step (until the prespecified number of features are selected).

Weaknesses
Empirical evaluation is not as extensive, however it is performed against relevant feature selection like competitor attention-based feature selection, as well as sequential LASSO and OMP (with neural networks).
It is encouraging that the regularized linear Sequential attention version had indistinguishable performance from the softmax based Sequential attention one, on benchmark data, but it is still not “bridging the gap”, but just leaves a chance that it might enjoy theoretical guarantees too.


**Summary Of The Paper:**

The paper is proposing a forward feature selection algorithm with the use of attention mechanism for assessing the feature relevance of currently not selected features. Since attention mechanism consider all unselected features at once, there is computational saving, compared to the common greedy forward selection algorithm. A specific instance of the Sequential attention feature selection framework, was shown to be equivalent to Orthogonal Matching Pursuit, thus inheriting theoretical guarantees on the quality of approximation.

**Summary Of The Review:**

Based on the perceived novelty and theoretical results, this paper seems like it could be accepted.

---

> ### Author Response · Authors · 2022-11-16
> **Thank you for the review**
>
> We thank Reviewer ABqs for endorsing our theoretical contributions (i.e., analyzing Sequential Attention via Sequential LASSO and OMP) and our novel use of attention to design an efficient and effective feature selection algorithm. We address some of the comments below:
>
> > Empirical evaluation is not as extensive, however it is performed against relevant feature selection like competitor attention-based feature selection, as well as sequential LASSO and OMP (with neural networks).
>
> As suggested by Reviewer vzVg, we added a comparison to the Concrete Autoencoder (CAE) algorithm [ABZ2019] (see Section 4.1 in our revised draft). For instance, on MNIST, our experiments show that Sequential Attention finds a set of $k = 50$ features with a prediction accuracy that is 5% better than CAE, in half the selection time. We are happy to take suggestions for additional evaluations that would be helpful in determining the quality of Sequential Attention.
>
> [ABZ2019] Muhammed Fatih Balın, Abubakar Abid, and James Zou. Concrete autoencoders: Differentiable feature selection and reconstruction. In International Conference on Machine Learning (ICML), pp. 444–453. PMLR, 2019.
>
> > It is encouraging that the regularized linear Sequential attention version had indistinguishable performance from the softmax based Sequential attention one, on benchmark data, but it is still not “bridging the gap”, but just leaves a chance that it might enjoy theoretical guarantees too.
>
> We agree that our work leaves open the possibility of a direct analysis of the softmax Sequential Attention algorithm as presented in Algorithm 1. We leave this as an exciting open problem for future works to explore.
>
> > Related work could have mentioned another relevant feature selection approach MRMR - Maximum Relevance Minimum Redundancy.
>
> We thank the reviewer for the reference to this information-theoretic method. We have added this reference to our discussion of prior work on feature selection. The MRMR algorithm is similar to the CMIM algorithm [Fle2004], which we have used as one of our baselines for our feature selection experiments on the large-scale Criteo dataset.
>
> [Fle2004] Francois Fleuret.  Fast binary feature selection with conditional mutual information. Journal of Machine Learning Research, 5(9), 2004.
>
> > Section 4.2 there is redundant “Algorithm Algorithm 1”
>
> Thank you – we corrected this in the revised draft.

---

### Author Response · Authors · 2022-11-12
**Initial Response to All Reviewers: Colab Notebook Demo Comparison (1/2)**

We thank the reviewers for their thoughtful and comprehensive reviews of our work.

In our initial response, we present a colab notebook (https://colab.research.google.com/drive/1uhcAkvpyqduVLUTk1jofi8IUq-32BbQI, created from an anonymous Google account) to demonstrate Sequential Attention and compare it to more baselines such as Liao–Latty–Yang 2021 (LLY) and Concrete Autoencoders (CAE) as suggested by Reviewer vzVg (in addition to the current baselines: Group LASSO, Sequential LASSO, OMP). We believe this notebook addresses several of the questions raised by the reviewers, and in this initial response, we address questions about the experiments which we believe are of interest to all of the reviewers. We will provide a second round of responses to each individual reviewer, but we felt it was appropriate to release this notebook ahead of time.

This notebook compares Sequential Attention, Group LASSO, Sequential LASSO, OMP, LLY, and CAE for feature selection where each method is given the same total epoch budget. For this number of epochs, we show that Sequential Attention gives the highest accuracy, closely followed by Sequential LASSO. We significantly outperform the other methods.

We respond to some of the questions raised that we believe (1) are of interest to all reviewers and (2) are addressed with this notebook. In particular, we respond to several questions raised by Reviewers tvep and vzVg concerning the experiments and comparisons to prior work.

## Reviewer tvep

> Finally, I believe that Algorithm 1, in its current form, is very inefficient since it requires training at least k models.

As noted in Appendix B.3 of our paper, we implement our algorithm to run during *one model training* by partitioning the training epochs into $k$ parts and selecting a feature in each of these $k$ parts. We provide the implementation of this one-pass version in the colab demo. In the revised version of this paper, we will provide a more detailed description of this one-pass version of Sequential Attention, Sequential LASSO, and OMP.

> How was the method implemented efficiently (as stated in the abstract)? What are the running speed and memory consumption? How do they compare to related approaches?

Our experiments in the submission and this colab notebook fix the total number of epochs used by each method (a proxy for resource costs and the running time since all model architectures are approximately the same size). Thus, our results highlight the quality of the algorithms and their criteria for feature selection when computational budgets remain constant.  We will provide additional experiments to give a fine-grained comparison about the running time cost of each method.

> However, I believe it has room for improvement, especially regarding the clarity of the experimental setup.

Our colab notebook presents all of the details of our experimental setup, including a full implementation of each algorithm, the choice of optimizer, hyperparameters, etc. We will provide a detailed explanation of the experimental setup and hyperparameter choices in a new subsection of the appendix.

---

> ### Author Response · Authors · 2022-11-12
> **Initial Response to All Reviewers: Colab Notebook Demo Comparison (2/2)**
>
> ## Reviewer vzVg
>
> > The algorithm is easy to reproduce. However, some hyper-parameters appear to be missing. The value for λ in the sequential LASSO is not explained. Some information regarding the training procedure (number of epochs, optimizer, ...) is also missing.
>
> We provide the full details of our feature selection training procedure and hyperparameters in our colab notebook demo. Regarding the choice of λ in Sequential LASSO and Group LASSO, we use one value of λ (`group_lasso_scale` in the colab notebook) as the regularization strength for all k steps of the sequential feature selection process. This λ parameter is then tuned to give the best overall feature selection results.
>
> > The experimental section is poor. The authors only compared their algorithm against other incremental FS algorithms. Although it is a good sign to establish similar or better results in this specific approach, there exists multiple FS algorithms with better results than the ones provided in the paper. There is a lack of comparison against similar FS architectures like CAE [1], DFS [2] or SFS [3].
>
>
> We provide an explicit comparison to CAE in our colab notebook (using the CAE implementation provided by the authors). The results show that Sequential Attention is superior to CAE for all `num_selected_features` considered. We note the DFS algorithm is quite similar to CAE, so the performance of CAE is likely representative since both methods seek to select a small subset of features in an unsupervised manner by learning a sparse autoencoder that depends on only a few features. We will add citations/discussions for these works and the CAE evaluations to our experiment figures in Section 4.
>
> We believe our new comparison to the LLY algorithm is representative of the SFS algorithm. Both algorithms are based on the idea of aggregating instance-wise feature importance scores (saliency) to give a global feature selection algorithm. The methods for obtaining this saliency are also similar — SFS uses a softmax score and LLY uses a similar attention-based score. As stated in our paper, we believe our main improvements over works such as LLY and SFS are (1) a simpler training process that directly trains a global feature importance score vector via a softmax mask instead of aggregating instance-wise information, and (2) our use of adaptivity to identify a *diverse set* of high-signal features.

---

### Decision · Program_Chairs · 2023-01-20

**Decision:**

Accept: poster

**Justification For Why Not Higher Score:**

All reviewers vote for acceptance, but without strong ratings.

**Justification For Why Not Lower Score:**

All reviewers vote for acceptance.

**Metareview: Summary, Strengths And Weaknesses:**

Summary:

This paper proposes a method to perform feature selection by learning to weigh the input features with probabilities given by a softmax transformation. Given the similarity to attention methods, the method is called Sequential Attention. It consists of running the model with the reminiscent non-selected features and then greedly selecting the features associated with the highest softmax weight. The authors provide a large body of theoretical guarantees, linking an adapted (regularized) version of Sequential Attention to the classical Sequential LASSO algorithm. Results on 7 datasets showcase the predictive performance of the proposed approach.

Strengths:

- Clearly written study, the methodology seems fairly novel.

- While equivalence of Sequential LASSO and OMP was shown under several assumptions before, this work shows the equivalence holds for more general settings. Clever use of attention to achieve computational savings by evaluating unselected features all at once, at every step (until the prespecified number of features are selected).

- The paper has a strong math foundation and excels at theoretical contributions.

- The proposed method outperforms related approaches despite being very simple.

- On top of this, this paper shows empirically and theoretically that "attention" is, in fact, a suitable approach to selecting a good subset of features.

- The paper is well written and organized.

- The paper proposes efficient sequential feature selection methods with attention, which reduces the complexity of greedy forward based feature selection.

- The authors provide provable guarantees for Sequential Attention for least squares linear regression by analyzing its variation called regularized linear Sequential Attention. They show the connection between Sequential LASSO and Orthogonal Matching Pursuit.

- The idea is easy to implement
- The theoretical explanation is interesting
- The spatial complexity of the algorithm is low

Weaknesses:

- Empirical evaluation is not as extensive, however it is performed against relevant feature selection like competitor attention-based feature selection, as well as sequential LASSO and OMP (with neural networks). It is encouraging that the regularized linear Sequential attention version had indistinguishable performance from the softmax based Sequential attention one, on benchmark data, but it is still not âbridging the gapâ, but just leaves a chance that it might enjoy theoretical guarantees too.

- My biggest concern in this paper is regarding clarity. The experimental details are often unspecified or hard to understand

- The authors claimed a lot of merits for the proposed algorithms, such as âthis technique reduces the overhead of our algorithm, removes the need to tune unnecessary hyperparameters, works directly with any model architecture,âŠachieves state-of-the-art feature selection results for neural networks on standard benchmarksâ. However, the experiments seem not quite complete to support the claim. For example, there is no experiment demonstrating the approach works on all model architectures.

- The advantages over existing methods are not clear or significant. The results in Figure 3 on DNNs seems to suggest there is limited advantage over SL, GL or OMP for these datasets.

- The scale of the experiment on DNN is quite small and limited. The experiments on neural networks are done with only a one-layer neural network with hidden width 67 and ReLU activation.

- The idea is somehow trivial
- The algorithm time complexity is extremely high
- The experimental results only cover other incremental Feature Selection methods.

Decision:

A maority of reviewers vote for acceptance. In the discussion, the only reviewer that voted for rejection said " I will consider to accept this paper if the rest of the reviewers vote for that.". Because of this, I, therefore, recommend accepting the paper and encourage the authors to use the feedback provided to improve the paper for its camera-ready version.

**Note From Pc:**

if the above contains the word "oral" or "spotlight" please see: "oral" presentation means -> notable-top-5% and "spotlight" means -> notable-top-25%. As stated in our emails, we are disassociating presentation type from AC recommendations

**Summary Of Ac-Reviewer Meeting:**

The only reviewer that voted for rejection said that he was happy to change his mind if all reviewers voted for acceptance.